# Architecture and evolution of the *cis*-regulatory system of the echinoderm *kirrelL* gene

**Jian Ming Khor, Charles A Ettensohn***

Department of Biological Sciences, Carnegie Mellon University, Pittsburgh, United States

**Abstract** The gene regulatory network (GRN) that underlies echinoderm skeletogenesis is a prominent model of GRN architecture and evolution. *KirrelL* is an essential downstream effector gene in this network and encodes an Ig-superfamily protein required for the fusion of skeletogenic cells and the formation of the skeleton. In this study, we dissected the transcriptional control region of the *kirrelL* gene of the purple sea urchin, *Strongylocentrotus purpuratus*. Using plasmid- and bacterial artificial chromosome-based transgenic reporter assays, we identified key *cis*-regulatory elements (CREs) and transcription factor inputs that regulate *Sp-kirrelL*, including direct, positive inputs from two key transcription factors in the skeletogenic GRN, Alx1 and Ets1. We next identified *kirrelL cis*-regulatory regions from seven other echinoderm species that together represent all classes within the phylum. By introducing these heterologous regulatory regions into developing sea urchin embryos we provide evidence of their remarkable conservation across ~500 million years of evolution. We dissected in detail the *kirrelL* regulatory region of the sea star, *Patiria miniata*, and demonstrated that it also receives direct inputs from Alx1 and Ets1. Our findings identify *kirrelL* as a component of the ancestral echinoderm skeletogenic GRN. They support the view that GRN subcircuits, including specific transcription factor–CRE interactions, can remain stable over vast periods of evolutionary history. Lastly, our analysis of *kirrelL* establishes direct linkages between a developmental GRN and an effector gene that controls a key morphogenetic cell behavior, cell–cell fusion, providing a paradigm for extending the explanatory power of GRNs.

**\*For correspondence:**
ettensohn@cmu.edu

**Competing interest:** The authors declare that no competing interests exist.

## Editor's evaluation

In this manuscript, Khor et al. examine the transcriptional regulation of kirrelL, a gene whose protein product is required for cell-cell fusion during the morphogenesis of the sea urchin larval skeleton. They establish a putative direct link between a developmental gene regulatory network driving cell fate commitment and an effector protein enabling a key behavior of the specified cell type, thereby strengthening the explanatory power of a well-established GRN model. It places a key morphoregulatory gene, kirrelL, into the extensively studied gene regulatory network of sea urchins and reveals deep evolutionary conservation of regulatory element function. This study should be of broad, general interest for developmental biologists.

## Introduction

Evolutionary changes in animal form have occurred through modifications to the developmental programs that give rise to anatomy. These developmental programs can be viewed as gene regulatory networks (GRNs), complex, dynamic networks of interacting regulatory (i.e., transcription factor-encoding) genes that determine the transcriptional states of embryonic cells (*Peter and Davidson,*

*2015*). Sea urchins and other echinoderms are prominent models for GRN biology for several reasons: (1) there are well-developed tools for dissecting developmental GRNs in these animals, (2) a large number of species that represent a wide range of evolutionary distances are amenable to study, and (3) there is a rich diversity of developmental modes and morphologies within the phylum (*Arnone et al., 2016*).

All adult echinoderms possess elaborate, calcified endoskeletons. Most species are maximal indirect developers; that is, they develop via a feeding larva that undergoes metamorphosis to produce the adult. The feeding larvae of echinoids (sea urchins) and ophiuroids (brittle stars) have extensive endoskeletons, holothuroids (sea cucumbers) have rudimentary skeletal elements, and asteroids (sea stars) lack larval skeletal elements entirely. Larval skeletons are thought to be derived within the echinoderms as the feeding larvae of hemichordates (acorn worms), the sister group to echinoderms, and the larvae of crinoids (sea lilies and feather stars), a basal echinoderm clade, lack skeletons. The skeletal cells of larval and adult echinoderms are similar in many respects, supporting the widely accepted view that the larval skeleton arose via co-option of the adult skeletogenic program (*Czarkwiani et al., 2013*; *Gao et al., 2015*; *Gao and Davidson, 2008*; *Killian et al., 2010*; *Mann et al., 2010*; *Mann et al., 2008*; *Richardson et al., 1989*).

The embryonic skeleton of euechinoid sea urchins, the best studied taxon, is formed by a specialized population of skeletogenic cells known as primary mesenchyme cells (PMCs). These cells are the progeny of the large micromeres (LMs), four cells that arise near the vegetal pole during early cleavage. The GRN that underlies PMC specification is one of the best characterized GRNs in any animal embryo (*Oliveri et al., 2008*; *Shashikant et al., 2018a*). This GRN is initially deployed through the activity of a localized maternal protein, Disheveled, which stabilizes β-catenin in the LM lineage, leading to the early zygotic expression of a repressor, *pmar1/micro1* (*Logan et al., 1999*; *Nishimura et al., 2004*; *Oliveri et al., 2002*; *Peng and Wikramanayake, 2013*; *Weitzel et al., 2004*). These molecular events lead to the zygotic expression of several regulatory genes selectively in the LM-PMC lineage. Two of the most important of these regulatory genes are *alx1* (*Ettensohn et al., 2003*) and *ets1* (*Kurokawa et al., 1999*), each of which is required for PMC specification and morphogenesis.

After their specification, PMCs undergo a spectacular sequence of morphogenetic behaviors that includes epithelial–mesenchymal transition (EMT), directional cell migration, cell fusion, and biomineral formation. PMCs undergo EMT at the late blastula stage, ingressing from the vegetal plate into the blastocoel. They migrate along the blastocoel wall and gradually arrange themselves in a ring-like pattern near the equator of the embryo. As they migrate, PMCs extend filopodia that fuse with those of neighboring PMCs, giving rise to a cable-like structure that joins the cells in a single, extensive syncytium. Beginning late in gastrulation and continuing throughout the remainder of embryogenesis, PMCs deposit calcified biomineral within the syncytial filopodial cable.

The complex sequence of PMC morphogenetic behaviors is regulated by hundreds of specialized effector proteins. The spatiotemporal expression patterns of these proteins are controlled by the GRN deployed in the LM-PMC lineage. A major current goal is to identify effector proteins that regulate specific PMC behaviors and elucidate the GRN circuitry that controls these genes (see *Ettensohn, 2013*; *Lyons et al., 2012*). Dissection of the *cis*-regulatory elements (CREs) that control essential morphogenetic effector genes, including the identification of specific transcription factor inputs, would directly link them to the relevant circuitry and provide a GRN-level explanation of developmental anatomy. At present, we have only a limited understanding of the *cis*-regulatory control of three PMC effector genes: two genes (*sm30* and *sm50*) that encode secreted proteins occluded in the biomineral (*Makabe et al., 1995*; *Walters et al., 2008*) and a third gene (*cyclophilin/cyp1*) of unknown function (*Amore and Davidson, 2006*).

KirrelL is a PMC-specific, Ig domain-containing, transmembrane protein required for cell–cell fusion (*Ettensohn and Dey, 2017*). The expression and function of the protein have been examined in two sea urchin species, *Strongylocentrotus purpuratus* and *Lytechinus variegatus*. In *kirrelL* morphants, PMCs extend filopodia and migrate but filopodial contacts do not result in fusion; this prevents the formation of the PMC syncytium and results in the secretion of small, unconnected biomineralized elements. In all echinoderms that have been examined, the *kirrelL* gene lacks introns, raising the possibility that its origin early in echinoderm evolution was a consequence of retrotransposition, a common gene transfer mechanism that results in intronless genes and one that has played a particularly prominent role in the diversification of Ig-domain-containing proteins (*Baertsch et al., 2008*; *Cordaux and*

*Batzer, 2009*; *Dermody et al., 2009*; *Farré et al., 2017*). The expression pattern of *kirrelL* is typical of many PMC effector genes . In *S. purpuratus*, *kirrelL* expression is first detectable at the blastula stage (~18 hpf) and peaks early in gastrulation (~30 hpf) (*Tu et al., 2014*). Expression then declines and is followed by a second peak at ~64 hpf, when *kirrelL* is expressed predominantly at sites of active skeletal rod growth as a consequence of localized, ectoderm-derived cues (*Sun and Ettensohn, 2014*). RNA-seq studies have shown that *Sp-kirrelL*, like many PMC effector genes, is positively regulated both by Alx1 and Ets1 (*Rafiq et al., 2014*). Although the gene has only been studied in detail in sea urchins, a recent study found that *kirrelL* is also expressed specifically in the embryonic skeletogenic mesenchyme of a brittle star, *Amphiura filiformis* (*Dylus et al., 2018*).

In the present study, we used plasmid- and bacterial artificial chromosome (BAC)-based transgenic reporter assays to identify key CREsand transcription factor inputs that regulate *kirrelL* in the sea urchin, *S. purpuratus*, directly linking this morphogenetic effector gene to the PMC GRN. In addition, we identified *kirrelL* cis-regulatory regions in echinoderm species from all major clades within the phylum and found that these regulatory regions drove PMC-specific expression in developing sea urchin embryos, highlighting their striking conservation across 450–500 million years of evolution. We analyzed in detail the *kirrelL* regulatory region of the sea star, *Patiria miniata*, and found that this gene, like *Sp-kirrelL*, receives direct inputs from Alx1 and Ets1. Our findings identify *kirrelL* as a component of the ancestral echinoderm skeletogenic GRN and strengthen the view that GRN subcircuits, including specific transcription factor–CRE interactions, can remain stable over very long periods of evolutionary history.

## Results

### The sea urchin *Sp-kirrelL* cis-regulatory landscape

We identified potential *Sp-kirrelL* CREs based on several criteria. We considered whether candidate regions were (1) hyperaccessible in PMCs relative to other cell types, (2) bound by Alx1, a key transcription factor in the PMC GRN and a positive regulator of *Sp-kirrelL*, (3) associated with active enhancer RNA (eRNA) expression, and (4) phylogenetically conserved at the level of DNA sequence. In a previous study, ATAC-seq and DNase-seq were used to identify regions of chromatin that are differentially accessible in PMCs relative to other cell types at the mesenchyme blastula stage (*Shashikant et al., 2018b*). ChIP-seq was used to identify binding sites of Sp-Alx1 at the same developmental stage (*Khor et al., 2019*). Recently, we used Cap Analysis of Gene Expression Sequencing (CAGE-seq) to profile eRNA expression at nine different stages of early sea urchin embryogenesis (*Khor et al., 2021*). Significantly, our integration of these different genome-wide datasets revealed several putative CREs located near *Sp-kirrelL*, some of which were found to share several signatures (*Figure 1A*). Developmental CAGE-seq profiles of eRNAs also provided additional information regarding temporal patterns of CRE activity (*Figure 1B*). To assist in identifying candidate CREs regulating the spatiotemporal expression of *Sp-kirrelL*, we used GenePalette (*Smith et al., 2017*) to perform phylogenetic footprinting of the *S. purpuratus* and *L. variegatus kirrelL* gene loci. Based on cross-species sequence conservation, cell type-specific DNA accessibility, Sp-Alx1-binding, and eRNA expression, we divided the intergenic sequences flanking *Sp-kirrelL* into nine putative CREs (labeled elements A–I) (*Figure 1C*). The elements were between 1.0 and 2.4 kb in size, with an average size of 1.5 kb.

### Characterization of functional *Sp-kirrelL* CREs

To test the transcriptional regulatory activity of candidate CREs (*Figure 2A*), we cloned them individually or in combination into the *EpGFPII* reporter plasmid, which contains a weak, basal sea urchin promoter, derived from the *Sp-endo16* gene, upstream of Green Fluorescent Protein (GFP) (see Materials and methods) and injected them into fertilized eggs. We observed that a GFP reporter construct containing upstream elements A–G recapitulated the correct spatial expression pattern of endogenous *Sp-kirrelL* with minimal ectopic expression (*Figure 2B, C* and *Figure 2—source data 1*). Further dissections revealed that a reporter construct containing elements D–G also drove strong GFP expression specifically in PMCs while a construct consisting of elements A–C showed weak GFP expression in PMCs. When elements were tested individually, we found that only elements C and G were able to drive GFP expression in sea urchin embryos. Element G, which is directly upstream of the *Sp-kirrelL* translational start site and contains part of the *Sp-kirrelL* 5' untranslated region (UTR), was

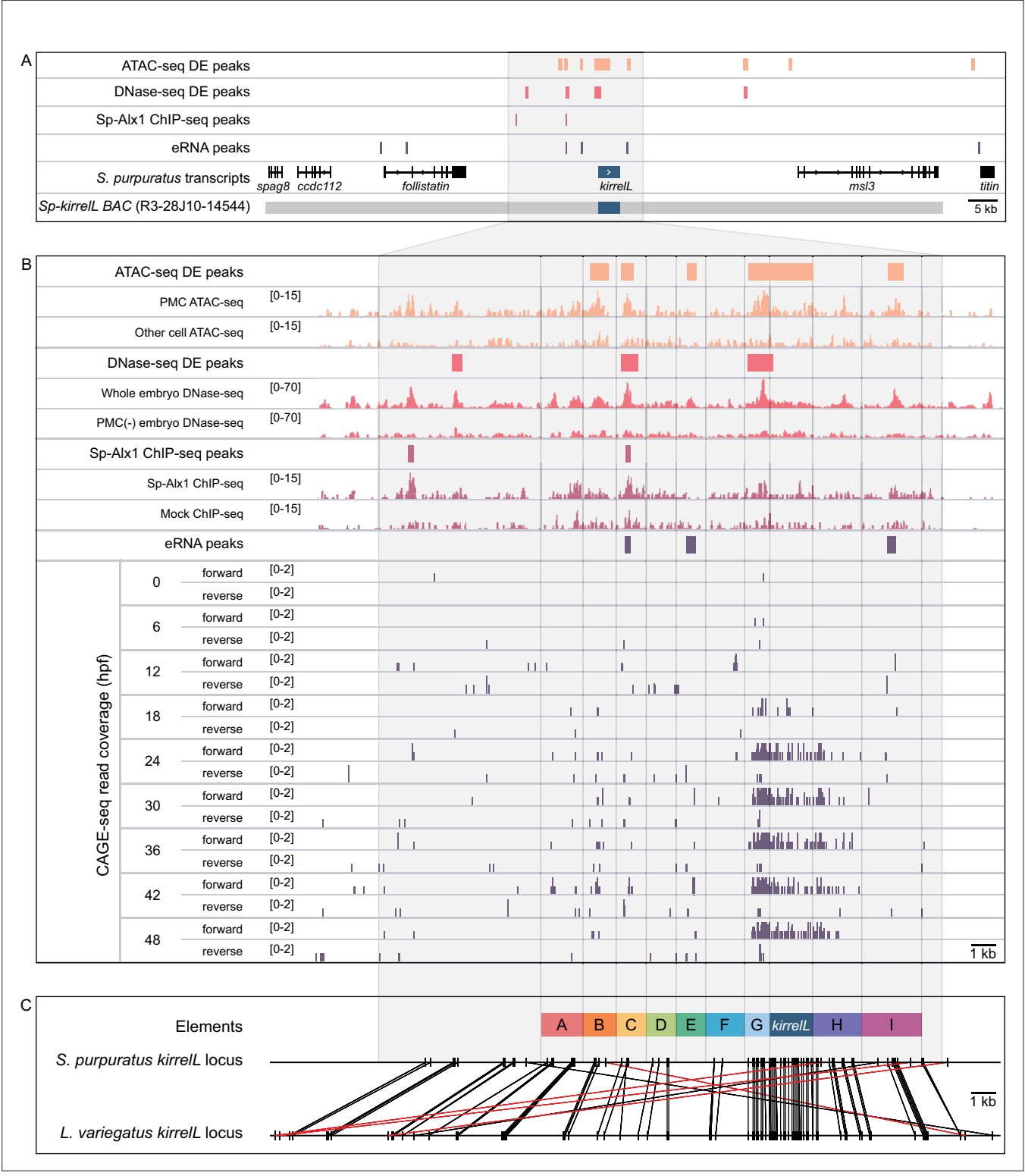

**Figure 1.** Characterization of the transcriptional regulatory landscape surrounding the *S. purpuratus kirrelL* (*Sp-kirrelL*) locus. (**A**) Diagram of the *Sp-kirrelL* locus showing neighboring genes, regions of chromatin differentially accessible in primary mesenchyme cells (PMCs) at the mesenchyme blastula stage (ATAC-seq DE peaks and DNase-seq DE peaks) (*Shashikant et al., 2018b*), Sp-Alx1-binding sites at the mesenchyme blastula stage (Sp-Alx1 ChIP-seq peaks) (*Khor et al., 2019*), and enhancer RNA (eRNA) peaks (union of all peaks from the nine developmental stages examined by *Khor et al., 2021*). (**B**) Signal obtained from each assay in the vicinity of the *Sp-kirrelL* locus. The bottom part of the panel shows the expression of eRNAs at the nine developmental stages analyzed by *Khor et al., 2021*. (**C**) Phylogenetic footprinting of genomic sequences near *S. purpuratus* and *L. variegatus*

*Figure 1 continued on next page*

Figure 1 continued

*kirrelL* (±10 kb of an exon) using GenePalette. Black lines indicate identical sequences of 15 bp or longer in the same orientation while red lines indicate identical sequences of 15 bp or longer in the opposite orientation. Nine putative *cis*-regulatory elements (CREs; labeled elements A–I) were identified based on sequence conservation and chromatin signatures.

observed to drive strong GFP expression specifically in the PMCs. Element C was also observed to drive GFP expression specifically in the PMCs, although fewer embryos expressed detectable levels of the reporter.

## Identification of direct transcriptional inputs into element C

We next focused on the molecular dissection of element C to identify direct transcriptional inputs into this CRE. Element C is noteworthy as it is differentially accessible in the PMCs based on both ATAC-seq and DNase-seq, bound by Sp-Alx1, and associated with eRNA expression (*Figure 3A*). We first performed a detailed dissection of element C to identify the minimal region that supported strong, PMC-specific GFP expression. We found that a reporter construct containing element C alone showed relatively weak reporter activity, similar to the construct containing elements A–C. In contrast, a larger, overlapping CRE we termed BC.ATAC, which included part, but not all, of element C, exhibited strikingly enhanced GFP expression in PMCs (*Figure 3—figure supplement 1A*). This difference in activity between element C and BC.ATAC suggested that element C might contain regulatory sites that have greater activity when in close proximity to the promoter.

To explore this further, we generated several reporter constructs consisting of truncated forms of element C, with boundaries defined by peaks from ATAC-seq (C.ATAC), DNase-seq (C.DNase), and Sp-Alx1 ChIP-seq (C.ChIP). The minimal element C region that showed strong, PMC-specific activity was determined to be C.ChIP. Increasing the distance between the C.ChIP element and the promoter (as in the C.DNase construct) significantly reduced enhancer activity. To predict transcription factor inputs within C.ChIP, we scanned the 200 bp C.ChIP sequence using JASPAR (*Mathelier et al., 2016*), with a focus on transcription factors known to be expressed at higher levels in PMCs than in other cell types. This analysis identified several candidate Alx1- and Ets1-binding sites (*Figure 3B* and *Figure 3—figure supplement 1B*). Consistent with previous RNA-seq analysis which has shown that *Sp-kirrelL* is sensitive to *alx1* and *ets1* knockdowns (*Rafiq et al., 2014*), our whole-mount in situ hybridization (WMISH) analysis of both *alx1* and *ets1* morphants confirmed that *Sp-kirrelL* expression declined to undetectable levels (*Figure 3—figure supplement 2*). Our mutational analysis of C.ChIP revealed that mutations of all putative Alx1- and/or Ets1-binding sites completely abolished GFP expression (*Figure 3C* and *Figure 3—figure supplement 1C*). In contrast, constructs containing mutations in putative Fox- or MEIS-binding sites exhibited reporter activity similar to that of the parental construct. Mutations of individual Alx1 and Ets1 sites revealed that Alx1 half site 2 and Ets1 site 1 provided key regulatory inputs.

## Analysis of the *Sp-kirrelL* promoter (element G)

To characterize the native *Sp-kirrelL* promoter region, we performed a detailed dissection of element G, which contains sequences directly upstream of the *Sp-kirrelL* translational start site, including the region encoding the *Sp-kirrelL* 5′-UTR (*Figure 4A*, *Figure 4—figure supplement 1A*, and *Figure 4—figure supplement 4A*). When tested in the *EpGFPII* plasmid, we found that a 301-bp region surrounding the transcriptional start site, a region we considered to include the *Sp-kirrelL* core promoter, drove ectopic GFP expression. As shown below, however, the same element failed to drive significant reporter expression in a BAC construct, indicating that the activity of the 310-bp element in *EpGFPII* was the result of abnormal synergy between the *Sp-kirrelL* and *Sp-endo16* promoters. We next performed mutational analysis of the minimal element G fragment that drove strongest PMC-specific GFP expression (G.ATAC). We determined that this CRE receives direct and positive inputs from Alx1 and Ets1, similar to the C.ChIP element (*Figure 4B*, *Figure 4—figure supplement 2*, and *Figure 4—figure supplement 4B*). For two constructs in which all Alx1- or all Ets1-binding sites were mutated, the difference in the numbers of embryos that exhibited PMC-specific versus ectopic expression as compared to the parental G.ATAC construct (see *Figure 2—source data 1*) was highly significant by a chi-square test (p < 0.001). Reporter constructs with mutated CEBPA-, Fos::Jun-, Fox-, MEIS-, or Tbrain-binding sites exhibited PMC-specific GFP expression similar to that of the parental

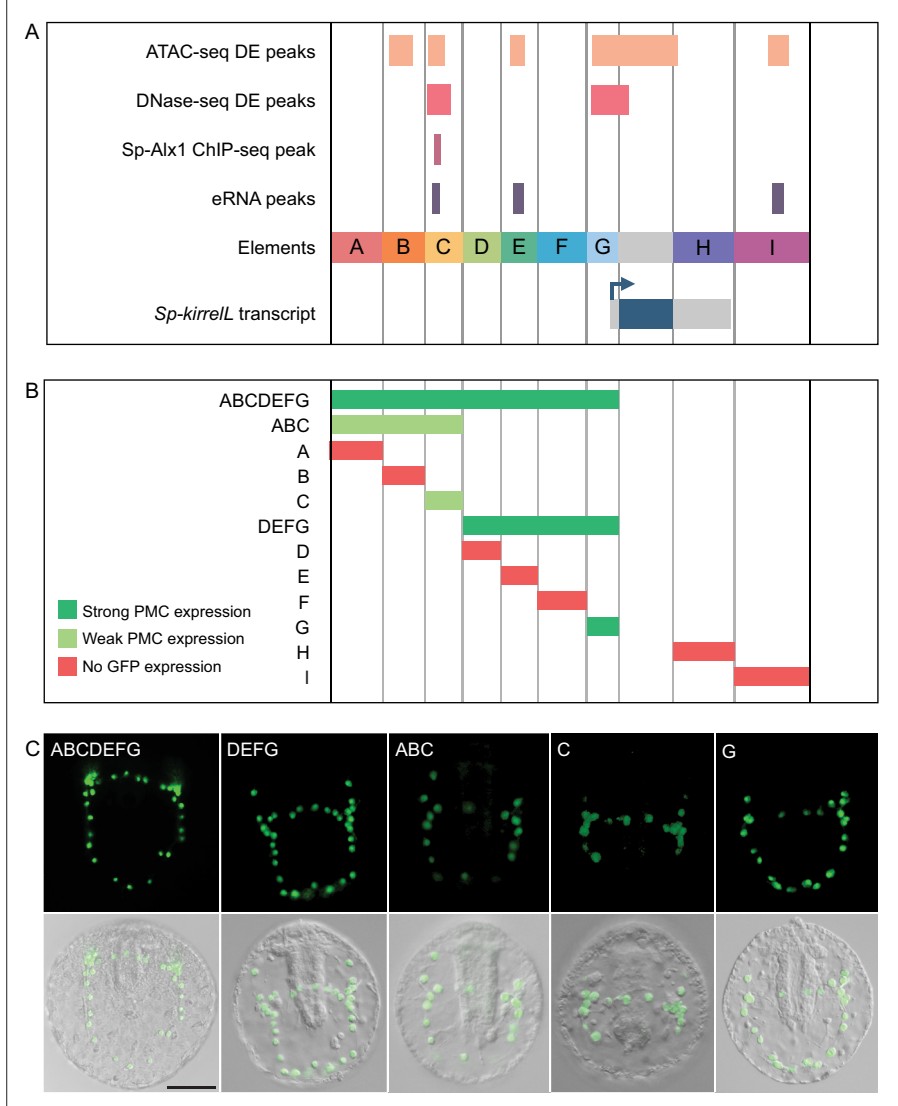

**Figure 2.** Functional analysis of noncoding genomic sequences flanking *Sp-kirrelL* to identify *cis*-regulatory elements (CREs). (**A**) Nine putative CREs (labeled elements A–I) were identified based on sequence conservation and previously published datasets (***Khor et al., 2021***; ***Khor et al., 2019***; ***Shashikant et al., 2018b***). (**B**) Summary of GFP expression regulated by putative CREs, as assessed by transgenic reporter assays. To be indicated as 'strong primary mesenchyme cell (PMC) expression', two criteria were satisfied: (1) more than 1/3 of all GFP-expressing embryos showed expression that was completely restricted to PMCs, and (2) the number of embryos in this class represented >15% of all injected embryos. 'Weak PMC expression' was defined similarly except that the number of embryos with expression completely restricted to PMCs represented <15% of all injected embryos. Complete scoring data for all constructs are contained in ***Figure 2—source data 1***. (**C**) Spatial expression patterns of GFP reporter constructs containing different *Sp-kirrelL* elements at 48 hr postfertilization (hpf). Top row: GFP fluorescence. Bottom row: GFP fluorescence overlayed onto differential interference contrast (DIC) images. Scale bar: 50 µm.

The online version of this article includes the following source data for figure 2:

**Source data 1.** Quantification of GFP expression patterns in embryos injected with reporter constructs.

construct. We also injected the different *Sp-kirrelL* element G truncations into fertilized *L. variegatus* eggs and observed similar expression patterns, indicating that inputs into element G are conserved in these two sea urchin species (***Figure 4—figure supplement 1B*** and ***Figure 4—figure supplement 4C***).

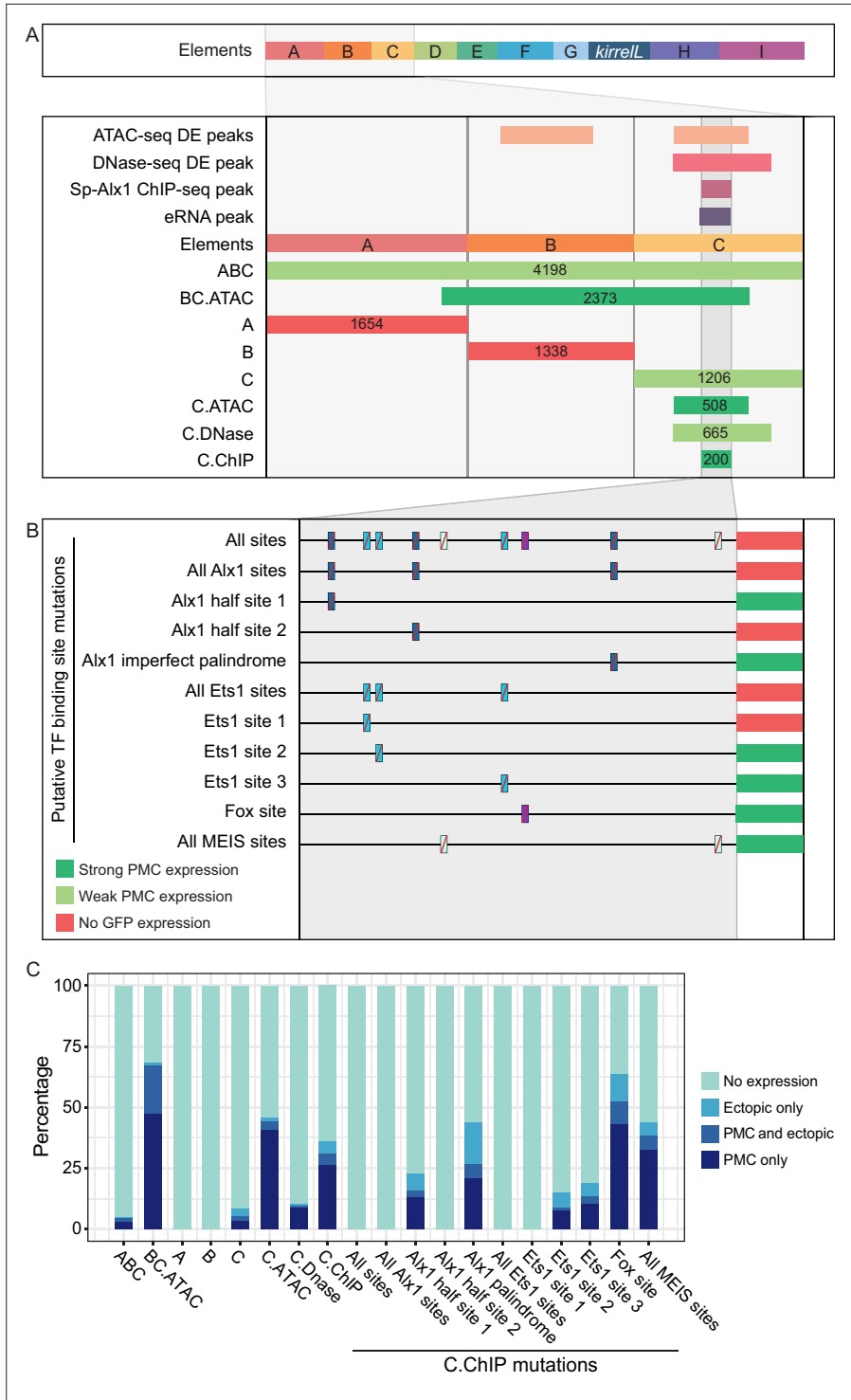

**Figure 3.** Molecular dissection of element C and the identification of direct transcriptional inputs. (**A**) Summary of transgenic GFP expression regulated by element C truncations using reporter constructs. Serial truncation of element C was performed based on boundaries of peaks defined by chromatin accessibility, Sp-Alx1-binding, and enhancer RNA (eRNA) expression. (**B**) Summary of GFP expression driven by C.ChIP element mutants. Criteria for strong and weak primary mesenchyme cell (PMC) expression are defined in *Figure 2*. (**C**) Stacked bar plot showing a summary of GFP expression patterns of injected embryos scored at 48 hpf. Each spatial expression category is expressed as a percentage of total injected embryos.

The online version of this article includes the following figure supplement(s) for figure 3:

*Figure 3 continued on next page*

*Figure 3 continued*

**Figure supplement 1.** Element C truncation and mutational analysis.

**Figure supplement 2.** Effects of *Sp-alx1* and *Sp-ets1* knockdown on *Sp-kirrelL* expression.

Our analysis of the native *Sp-kirrelL* promoter prompted us to investigate whether the addition of this region to our *EpGFPII* reporter constructs would allow us to uncover interactions between CREs and the native promoter that would have otherwise been missed. Strikingly, we found that elements B, C, E, F, H, and I were individually able to drive strong PMC-specific GFP expression when cloned adjacent to the *Sp-kirrelL* promoter, although these elements had previously exhibited minimal activity in the context of the *Sp-endo16* promoter alone (*Figure 4C*, *Figure 4—figure supplement 3A, B*, and *Figure 4—figure supplement 4D*; compared to *Figure 2*). As an example, element C in combination with the *Sp-endo16* promoter drove GFP expression in only 8.6% of embryos, the *Sp-kirrelL* promoter region in combination with the *Sp-endo16* promoter drove GFP expression in 19% of the embryos, but the combination of element C with the two promoter elements drove expression in 37.8% of embryos (*Figure 2—source data 1*). The elevated expression of the latter construct indicated that the its activity was not due solely to the additive activity of the C and SpkirrelL promoter elements interacting independently with the *Sp-endo16* promoter (37.8 > 8.6 + 19). In addition, we found that the C element exhibited substantial activity in the context of the *Sp-kirrelL* promoter alone, in the absence of the *Sp-endo16* promoter (*Figure 4—figure supplement 3*). We also observed that the presence of the native *Sp-kirrelL* promoter mitigated the need for the C.ChIP element within element C to be adjacent to the promoter for strong PMC-specific GFP expression. We confirmed that enhancer activity was dependent on the sequence of the *Sp-kirrelL* promoter, as GFP expression was abolished in a construct where the sequence was shuffled (*Figure 4—figure supplement 3C*).

To test whether the effect of deleting the region between C.ChIP and the promoter was due to the removal of repressor sites or to a change in the spacing between C.ChIP and the promoter, we generated and tested a construct that contained the region in question but in which the sequence of that region was randomly scrambled (*Figure 4—figure supplement 3D, E* and *Figure 4—figure supplement 4E*). We found that insertion of this sequence decreased activity compared to when C.ChIP was directly adjacent to the promoter. This supports the view that the principle effect of deleting this region was to decrease the spacing between C.Chip and the promoter rather than removing repressor sites. Taken together, these findings showed that several CREs are capable of functioning in concert specifically with the native *Sp-kirrelL* promoter and that this can bypass spacing hurdles that are evident when the *Sp-endo16* promoter alone is present.

## Relative contributions of individual CREs in the context of the entire *Sp-kirrelL* regulatory apparatus

Our analysis identified multiple CREs in the vicinity of the *Sp-kirrelL* locus that were capable of driving PMC-specific reporter expression when cloned into plasmids that contained the endogenous *Sp-kirrelL* promoter. To explore further the relative contributions of these various elements to *Sp-kirrelL* expression in vivo, we examined their function in the context of the complete transcriptional control system of the gene. For these studies, we utilized a 130-kb BAC that contained the single exon *Sp-kirrelL* gene, flanked by 65 kb of sequences in each direction. We used recombination-mediated genetic engineering (recombineering) to replace the single *Sp-kirrelL* exon seamlessly with either GFP or mCherry coding sequence (*Figure 5A*). We found that *Sp.kirrelL.mCherry.BAC* faithfully recapitulated the expression of endogenous *Sp-kirrelL* in the PMCs at 48 hpf with minimal ectopic expression (*Figure 5—figure supplement 1*). We next generated deletion mutants based on results from our plasmid GFP reporter assays to quantitatively assess the contributions of elements A–G to *Sp-kirrelL* transcriptional regulation. We found that deletion of elements A–G (ΔCRE.GFP.BAC) completely abolished GFP expression. We also observed that retaining the minimal endogenous *Sp-kirrelL* promoter (ΔCRE.kirrelLprm.GFP.BAC) did not rescue GFP expression, demonstrating that elements A–G are necessary for PMC-specific *Sp-kirrelL* expression in the context of the *Sp.kirrelL.GFP.BAC* consistent with our previous, plasmid-based analysis.

To directly compare the spatial expression patterns of deletion mutants with that of the parental BAC, we generated BAC mutants containing deletion of individual elements and coinjected them into

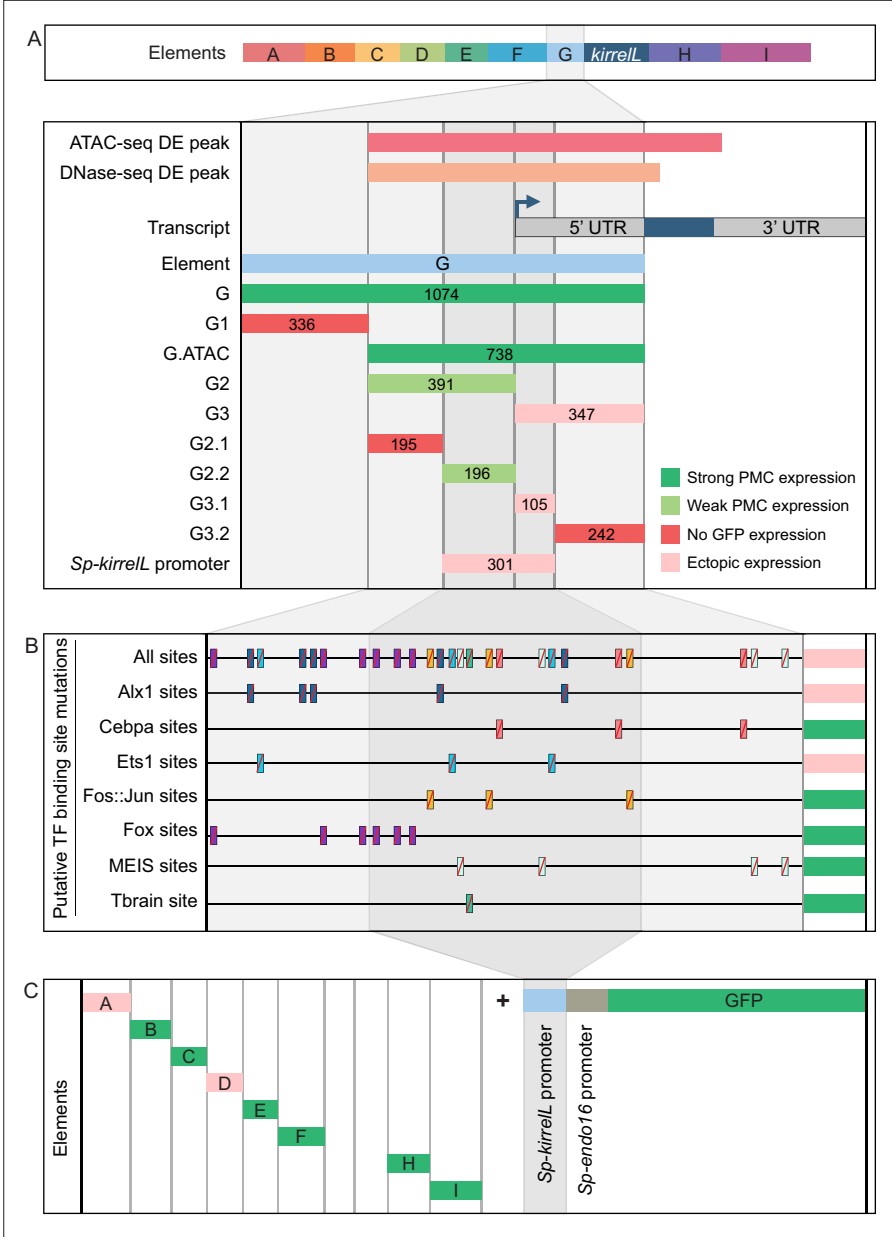

**Figure 4.** Molecular dissection and mutation of element G. (**A**) Summary of GFP expression regulated by element G truncations using *EpGFPII* reporter constructs. Serial truncation of element G was performed based on boundaries defined by chromatin accessibility and the *kirrelL* 5'-UTR. Criteria for strong and weak primary mesenchyme cell (PMC) expression are defined in *Figure 2*. Ectopic expression is defined as majority of injected embryos exhibiting GFP expression in cells other than PMCs. (**B**) Summary of GFP expression driven by G.ATAC element mutants using *EpGFPII* reporter constructs. (**C**) Analysis of element enhancer activity in modified *EpGFPII* reporter constructs containing the endogenous *Sp-kirrelL* promoter elements.

The online version of this article includes the following figure supplement(s) for figure 4:

**Figure supplement 1.** Element G truncation and mutational analysis.

**Figure supplement 2.** Mutational analysis of G.ATAC element.

**Figure supplement 3.** Interactions between *Sp-kirrelL cis*-regulatory elements (CREs) and the endogenous *Sp-kirrelL* promoter.

**Figure supplement 4.** Stacked bar plots showing summary of GFP expression patterns of injected embryos scored at 48 hpf.

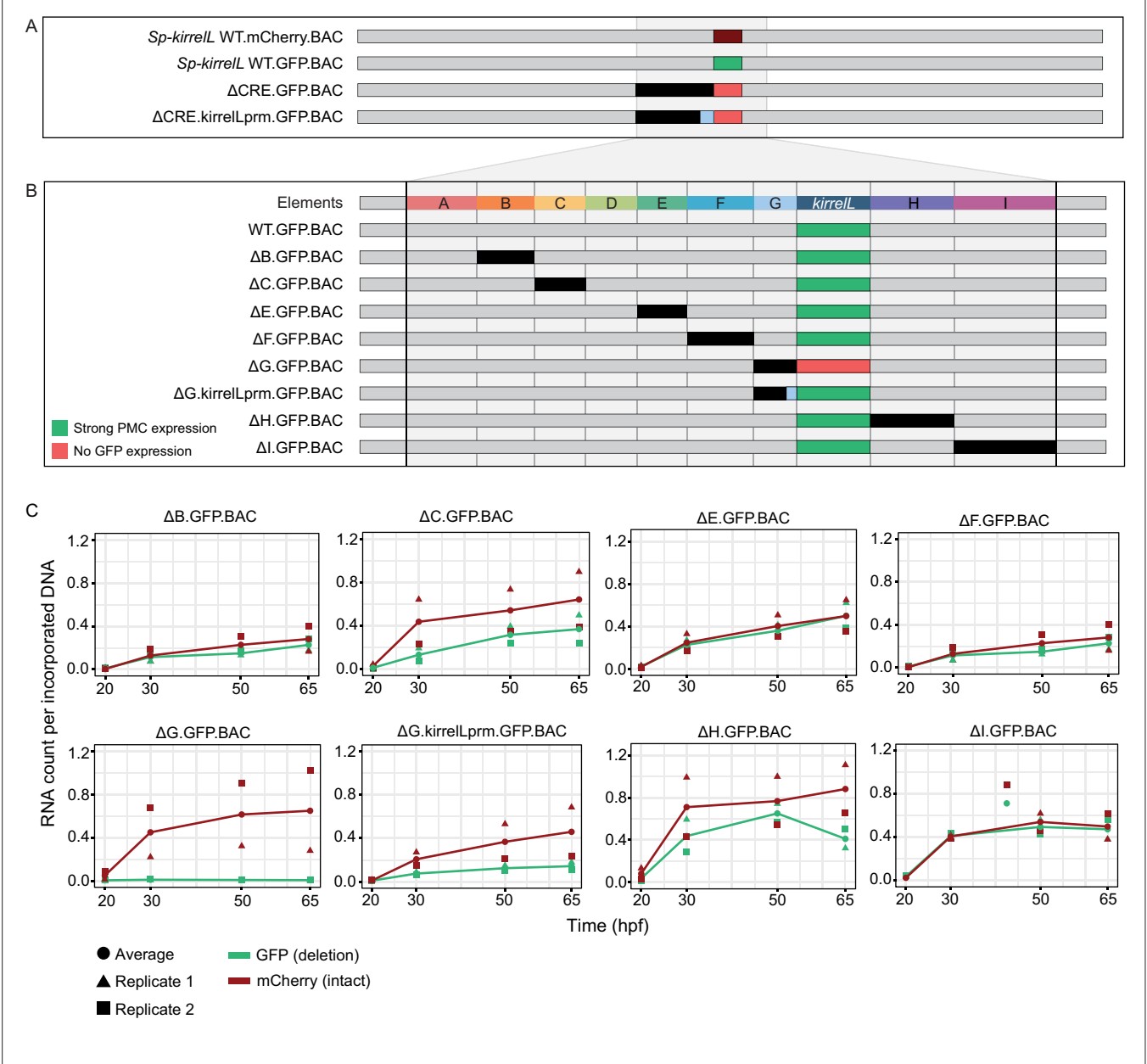

**Figure 5.** *Sp-kirrelL cis*-regulatory analysis using BACs. (**A**) BAC deletions show that elements A–G are necessary for GFP expression, regardless of the presence of the endogenous *Sp-kirrelL* core promoter elements. (**B**) Summary of GFP expression patterns of individual *Sp-kirrelL* elements using GFP BAC deletions. Criteria for strong primary mesenchyme cell (PMC) expression are defined in ***Figure 2***. (**C**) Quantitative NanoString analysis of reporter expression in embryos coinjected with parental mCherry and mutant GFP BACs. Embryos were collected at 20, 30, 50, and 65 hpf. The average expression profile for each pair of BAC injection was calculated from NanoString counts of two biological replicates (see Materials and methods).

The online version of this article includes the following source data and figure supplement(s) for figure 5:

**Source data 1.** Summary of NanoString analysis.

**Source data 2.** Raw NanoString data for BAC-injected embryos.

**Source data 3.** NanoString analysis probe target sequences.

**Figure supplement 1.** Spatial expression patterns of embryos coinjected with parental mCherry and mutant GFP BACs.

**Figure supplement 2.** Stacked bar plots showing summary of GFP expression patterns of transgenic embryos injected with *Sp-kirrelL* GFP BACs containing different element deletions.

fertilized eggs with a parental mCherry BAC. We found that a BAC containing deletion of the element G (ΔG.GFP.BAC, which included a deletion of the *Sp-kirrelL* promoter) abolished GFP expression at 48 hpf (*Figure 5B*, *Figure 5—figure supplement 2*). By contrast, deletion of all of element G except for the promoter region (ΔG.kirrelLpromoter.GFP.BAC), resulted in a GFP spatial expression pattern similar to that of the parental mCherry. These findings confirmed the importance of the *Sp-kirrelL* promoter in supporting PMC-specific expression of the gene and showed that this region is essential even when all distal CREs are present. BACs containing individual deletions of other elements all remained active at 48 hpf and supported PMC-specific reporter expression, pointing to considerable redundancy in the contribution of each element to *Sp-kirrelL* expression.

To examine the relative contribution of distal CREs more rigorously, we measured levels of reporter transcripts using a NanoString nCounter. For each mutant BAC, we coinjected embryos with mCherry tagged, parental BAC and the GFP-tagged, mutant BAC and quantified the expression level of each reporter gene at four time points (20, 30, 50, and 65 hpf) (*Figure 5C* and *Figure 5—source data 1*). We found that deletion of element C resulted in approximately a 50% reduction in expression compared to WT BAC. As we observed previously, GFP expression was completely abolished when element G was deleted (ΔG.GFP.BAC) and this effect was diminished when the *Sp-kirrelL* promoter was retained (ΔG.kirrelLprm.GFP.BAC). Quantitative analysis revealed, however, that retention of the *Sp-kirrelL* promoter alone resulted in only a partial rescue of expression, with overall levels reduced substantially compared to the wild-type BAC. We also observed that deletion of element H resulted in decreased expression levels. Deletions of elements A and D were not tested as there was no evidence from our plasmid reporter analysis that these were functional CREs. Taken together, our qualitative and quantitative analyses show that at early stages of embryo development, *Sp-kirrelL* expression is controlled by multiple CREs, notably the C, G, and H modules, acting in concert with the *Sp-kirrelL* promoter.

## Cross-species analysis of echinoderm *kirrelL* CREs

As the noncoding region directly upstream of the translational start site of *Sp-kirrelL* was found to contain transcriptional control elements, we asked whether sequences upstream of *kirrelL* genes from other echinoderm classes might contain functionally conserved CREs that have activity in *S. purpuratus* PMCs. To date, the embryonic expression of *kirrelL* has been examined in two sea urchins (*S. purpuratus* and *L. variegatus*) and a brittle star (*A. filiformis*) (*Dylus et al., 2018*; *Ettensohn and Dey, 2017*); in all three species, embryonic expression is restricted to skeletogenic mesenchyme cells. We cloned ~1- to 2-kb noncoding sequences (see *Figure 6—source data 1* and *Figure 6—source data 2*) directly upstream of the translational start sites of *kirrelL* genes from *Eucidaris tribuloides* (pencil urchin), *Parastichopus parvimensis* (sea cucumber), *P. miniata* (sea star), *Acanthaster planci* (crown-of-thorns starfish), *Ophionereis fasciata* (brittle star), and *Anneissia japonica* (feather star) into the *EpGFPII* plasmid and injected them into fertilized *S. purpuratus* eggs (*Figure 6A*, *Figure 6—figure supplement 1*, and *Figure 6—figure supplement 2A*). Remarkably, we found that all six drove GFP expression in sea urchin embryos, with five out of six exhibiting strong GFP expression selectively in PMCs (*Figure 6B* and *Figure 6—figure supplement 3*). Taken together, these observations indicate that *kirrelL* CREs across echinoderm species are highly conserved in function. We found it particularly striking that *kirrelL* CREs from deeply divergent echinoderm species that do not form embryonic or larval skeletons (sea stars and feather stars) drive PMC-selective GFP expression in sea urchin embryos.

Although KirrelL has been shown to be an important morphoeffector gene in the sea urchin embryo, where it plays an essential role in PMC–PMC fusion, its expression in adult sea urchins has not been examined. We observed *Lv-kirrelL* expression in the skeletogenic centers of the adult rudiment and in the spine of 5-week-old juvenile sea urchins (*Figure 6C*). The expression pattern of *Lv-kirrelL* was very similar to that of *Lv-msp130r2*, a highly expressed biomineralization gene (*Figure 6—figure supplement 2B*). In contrast, expression of *Pm-kirrelL* was not detected during early embryonic and larval development in *P. miniata*, which does not from a larval skeleton (*Figure 6D*). *Pm-kirrelL* is, however, expressed in the developing adult rudiment in premetamorphic, late-stage sea star larva and in the adult skeletogenic centers in juvenile sea stars (*Figure 6D*). As a control, we showed *Pm-ets1* expression in the mesenchyme cells during early development and an expression pattern in the adult rudiment and skeletogenic centers in juvenile sea stars that closely resembled that of *Pm-kirrelL* (*Figure 6—figure supplement 2C*).

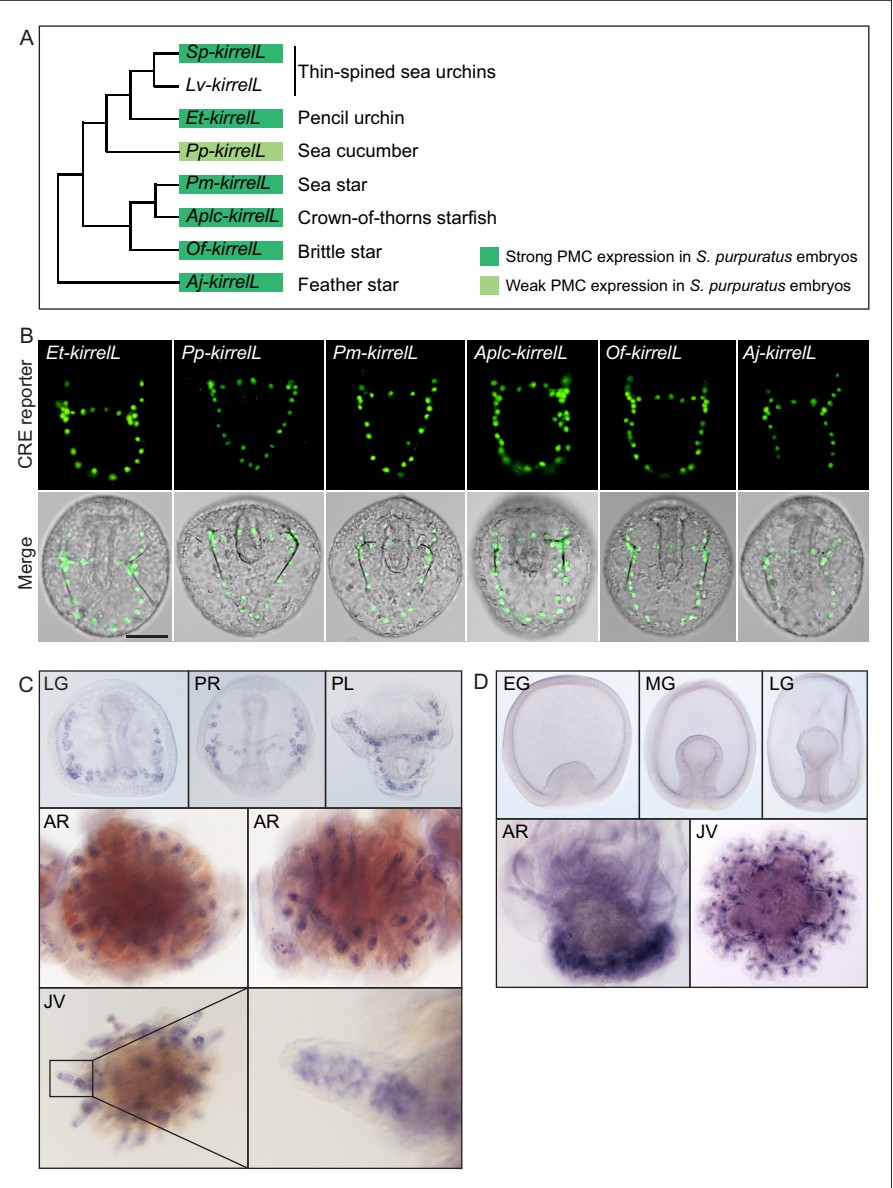

**Figure 6.** Cross-species analysis of *kirrelL cis*-regulatory elements (CREs) from diverse members of the echinoderm phylum. (**A**) Phylogenetic relationships of *kirrelL* genes based on the consensus view of evolutionary relationships among echinoderms. Branch lengths are not drawn to scale. Box colors correspond to expression of GFP in *S. purpuratus* embryos, driven by noncoding sequences upstream of *kirrelL* genes of *Eucidaris tribuloides* (*Et-kirrelL*), *Parastichopus parvimensis* (*Pp-kirrelL*), *Patiria miniata* (*Pm-kirrelL*), *Acanthaster planci* (*Aplc-kirrelL*), *Ophionereis fasciata* (*Of-kirrelL*), and *Anneissia japonica* (*Aj-kirrelL*). Criteria for strong and weak primary mesenchyme cell (PMC) expression are defined in *Figure 2*. (**B**) Spatial expression patterns of GFP reporter constructs containing *kirrelL* CREs from other echinoderm species in *S. purpuratus* embryos at 48 hpf. Top row: GFP fluorescence. Bottom row: GFP fluorescence overlayed onto differential interference contrast (DIC) images. Scale bar: 50 µm. (**C**) Representative whole-mount in situ hybridization (WMISH) images showing *Lv-kirrelL* expression during *L. variegatus* development. (**D**) *Pm-kirrelL* expression during *P. miniata* development. EG, early gastrula; MG, midgastrula; LG, late gastrula; PR, prism stage; PL, pluteus larva; AR, adult rudiment; JV, juvenile stage. All genomic coordinates and DNA sequences for the CREs are shown in *Figure 6—source data 2*.

The online version of this article includes the following source data and figure supplement(s) for figure 6:

**Source data 1.** Sequence coordinates for echinoderm *kirrelL cis*-regulatory elements (CREs) tested (*Arshinoff et al., 2022*; *Long et al., 2016*).

**Source data 2.** DNA sequences for *cis*-regulatory elements (CREs) validated in this study from *Eucidaris*

*Figure 6 continued on next page*

*Figure 6 continued*

*tribuloides* (*Et-kirrelL*), *Parastichopus parvimensis* (*Pp-kirrelL*), *Patiria miniata* (*Pm-kirrelL*), *Acanthaster planci* (*Aplc-kirrelL*), *Ophionereis fasciata* (*Of-kirrelL*), and *Anneissia japonica* (*Aj-kirrelL*).

**Source data 3.** Echinoderm primary mesenchyme cell (PMC)-specific Ig-domain protein sequences from *Strongylocentrotus purpuratus* (Sp), *Lytechinus variegatus* (Lv), *Eucidaris tribuloides* (Et), *Parastichopus parvimensis* (Pp), *Patiria miniata* (Pm), *Acanthaster planci* (Aplc), *Ophionereis fasciata* (Of), and *Anneissia japonica* (Aj) used for tree construction.

**Figure supplement 1.** Unrooted, maximum likelihood tree showing clustering of primary mesenchyme cell (PMC)-specific Ig-domain proteins from different echinoderm species that represent all classes within the phylum.

**Figure supplement 2.** Alignment of echinoderm KirrelL proteins and representative whole-mount in situ hybridization (WMISH) images of positive control probes.

**Figure supplement 3.** Stacked bar plots showing summary of GFP expression patterns of transgenic *S. purpuratus* embryos injected with constructs containing noncoding sequences upstream of *kirrelL* genes of *Eucidaris tribuloides* (*Et-kirrelL*), *Parastichopus parvimensis* (*Pp-kirrelL*), *Patiria miniata* (*Pm-kirrelL*), *Acanthaster planci* (*Aplc-kirrelL*), *Ophionereis fasciata* (*Of-kirrelL*), and *Anneissia japonica* (*Aj-kirrelL*).

## Dissection of a candidate adult skeletogenic CRE

As sea stars do not form a larval skeleton but express *kirrelL* specifically in adult skeletogenic centers, we exploited the activity of the *Pm-kirrelL* CRE in sea urchin embryos as a potential proxy for identifying transcriptional inputs that ordinarily control this gene in adult echinoderms (see Discussion). We performed truncations and mutations of the regulatory regions upstream of the *Pm-kirrelL* gene to identify direct transcriptional inputs (**Figure 7A and B**). Subdivision of the ~4 kb Pm-kirrelL regulatory region showed that activity was restricted to the proximal region (Pm2), and further analysis revealed that a 614-bp region (PmG) was sufficient to drive strong PMC-specific GFP expression in *S. purpuratus* embryos (**Figure 7B**, **Figure 7—figure supplement 1A**, and **Figure 7—figure supplement 2A**). Phylogenetic footprinting of genomic sequences from *P. miniata* and the closely related crown-of-thorns starfish (*A. planci*) showed substantial similarity in this region (**Figure 7A**). We performed mutational analysis of the PmG element and found that this CRE receives positive inputs from both Alx1 and Ets1 (**Figure 7C**, **Figure 7—figure supplement 1B**, **Figure 7—figure supplement 2B**, and **Figure 7—figure supplement 3A, B**), similar to the *Sp-kirrelL* C and G.ATAC elements. For two constructs in which all Alx1- or all Ets1-binding sites were mutated, the difference in the numbers of embryos that exhibited PMC-specific versus ectopic expression as compared to the parental PmG construct (see **Figure 2—source data 1**) was highly significant by a chi-square test ($p < 0.001$).

We next asked whether PmG1 and PmG2 elements, which are located near the *Pm-kirrelL* transcriptional start site, could interact with distal *Sp-kirrelL* elements, thereby substituting for the endogenous *Sp-kirrelL* promoter. For this analysis, we generated chimeric *EpGFPII* reporter constructs that contained the sea urchin *Sp-kirrelL* element C (SpC) adjacent to the sea star PmG1 or PmG2 element (**Figure 7D**). We found that PmG1 and PmG2 were both interchangeable with the *Sp-kirrelL* promoter and that interactions between SpC and PmG1 or PmG2 supported strong PMC-specific GFP expression in *S. purpuratus* embryos (**Figure 7—figure supplement 1C** and **Figure 7—figure supplement 2C**). PmG1 and PmG2 each conferred a roughly similar increase in expression frequency and specificity to element C as the *Sp-kirrelL* promoter region (**Figure 2—source data 1**). In a construct containing a PmG2 element with shuffled sequence, GFP expression was abolished. Additionally, we examined the effects of *Sp-alx1* and *Sp-ets1* knockdown on the activity of the *P. miniata* regulatory region and the *S. purpuratus* C and G elements (**Figure 7—figure supplement 4**). We confirmed that knockdown of Alx1 or Ets1 expression substantially suppresses the activity of all constructs in PMCs. These observations highlight a striking conservation of sequence and function in *kirrelL* promoters from deeply divergent echinoderm species.

## Discussion
### Linking developmental GRNs to morphogenesis

Recent studies with echinoderms have elucidated the architecture of developmental GRNs, including the GRN deployed specifically in embryonic skeletogenic mesenchyme of sea urchins (***Shashikant***

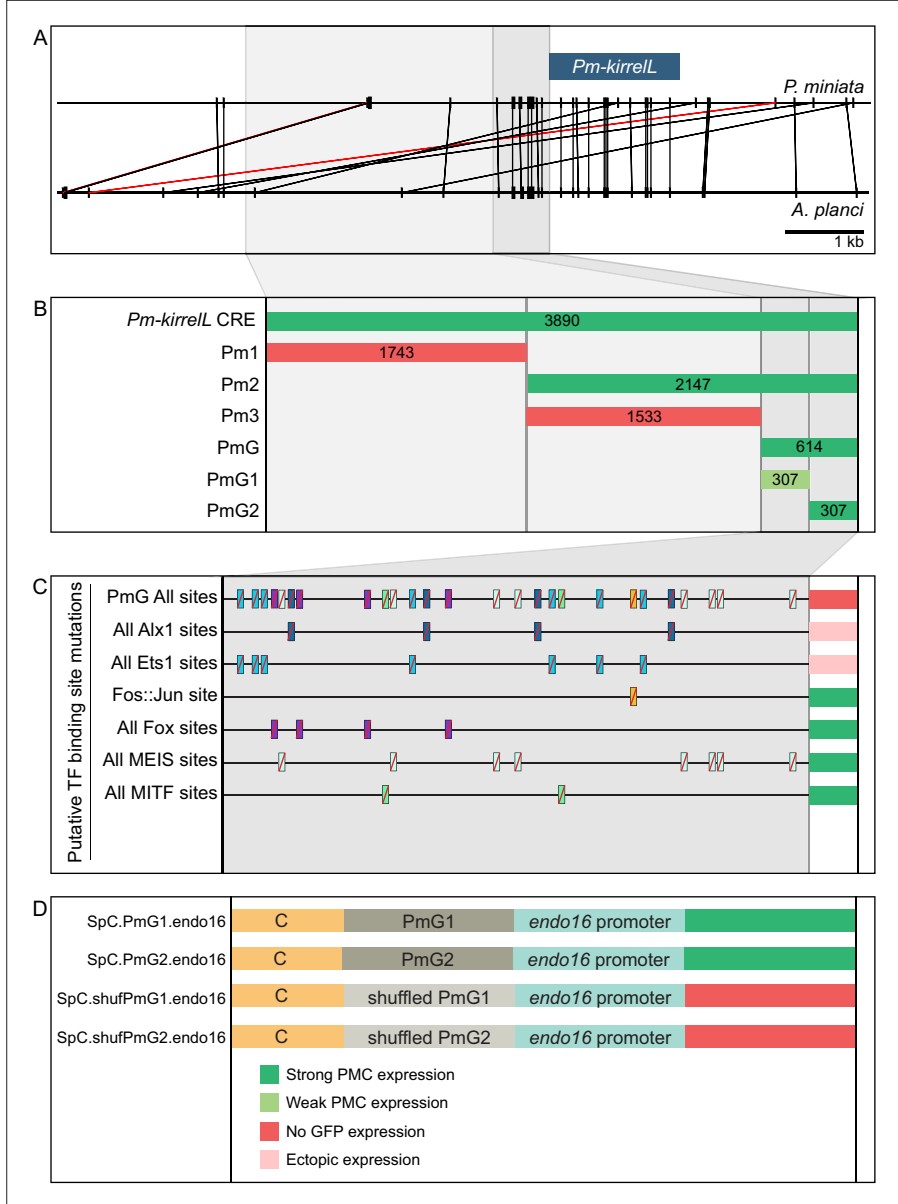

**Figure 7.** Functional analysis of noncoding genomic sequences upstream of *Pm-kirrelL* to identify *cis*-regulatory elements (CREs). (**A**) Phylogenetic footprinting of genomic sequences near *P. miniata* and *A. planci* kirrelL using GenePalette. Black lines indicate identical sequences of 13 bp or longer in the same orientation while red lines indicate identical sequences of 13 bp or longer in the opposite orientation. (**B**) Summary of GFP expression regulated by noncoding sequences upstream of the *Pm-kirrelL* translational start site. (**C**) Summary of GFP expression driven by PmG element mutants. (**D**) Summary of GFP expression regulated by chimeric reporter constructs containing *Sp-kirrelL* element C and *Pm-kirrelL* G1 or G2 elements. Criteria for strong and weak primary mesenchyme cell (PMC) expression are defined in *Figure 2*. Ectopic expression is defined as majority of injected embryos exhibiting GFP expression in cells other than PMCs.

The online version of this article includes the following figure supplement(s) for figure 7:

**Figure supplement 1.** Sea star *Pm-kirrelL cis*-regulatory element (CRE) truncation and mutational analysis.

**Figure supplement 2.** Stacked bar plots showing summary of GFP expression patterns of injected embryos scored at 48 hpf.

**Figure supplement 3.** Mutational analysis of the sea star *P. miniata kirrelL* element G (PmG).

**Figure supplement 4.** Effects of *Sp-alx1* and *Sp-ets1* knockdown on transgenic *Sp-kirrelL cis*-regulatory element (CRE) reporter construct expression.

*et al., 2018a*). Although these studies have focused largely on interactions among regulatory genes that constitute the core of such networks, the importance of GRNs from a developmental perspective is that they underlie the dramatic anatomical changes that characterize embryogenesis (*Ettensohn, 2013*; *Smith et al., 2018*). In that respect, GRNs have considerable power in explaining the transformation of genotype into phenotype. Moreover, if GRNs are to be useful in understanding the evolution of morphology, currently a major goal of comparative GRN biology, the developmental mechanisms by which these genetic networks drive morphology must be addressed. This work seeks to partially fill this conceptual gap by elucidating the transcriptional control of *Sp-kirrelL*, an effector gene required for cell–cell fusion, an important morphogenetic behavior of PMCs.

## The *cis*-regulatory apparatus of *Sp-kirrelL*

The combinatorial control of CRE function is important for driving complex gene expression patterns during animal development. In the present study, we identified key regulatory elements and transcription factor inputs that control *Sp-kirrelL* expression. Using plasmid reporter constructs, we identified seven CREs (elements B, C, E, F, G, H, and I) that were individually sufficient to drive strong PMC-specific GFP expression when placed adjacent to the native *Sp-kirrelL* promoter. Most of these same elements failed to drive reporter expression at detectable levels, however, when cloned directly adjacent to the 140-bp *Sp-endo16* core promoter, a component of *EpGFPII*, a vector widely used for *cis*-regulatory analysis in sea urchins. As proximal promoter elements have been shown to tether more distal elements in other organisms (*Calhoun et al., 2002*), we hypothesize that such tethering activity is present in the 301-bp *Sp-kirrelL* promoter element contained in element G. Tethering activity would also account for the fact the regulatory sites in the C element (i.e., those contained in C.ATAC and C.ChIP) must be in close proximity to the *Sp-endo16* promoter to activate transcription, while these same sites can function at a greater distance when working in concert with the endogenous *Sp-kirrelL* promoter. These findings highlight the potential limitations of transgenic reporter assays that rely exclusively on exogenous and/or core promoters.

As multiple CREs were capable of supporting PMC-specific reporter expression in combination with the *Sp-kirrelL* promoter, we performed BAC deletion analysis to determine the relative contributions of these elements to *Sp-kirrelL* expression. We quantified reporter expression using a newly developed, Nanostring-based assay that allowed us to measure the extent of transgene incorporation and reporter expression. We found that a single deletion of elements A–G entirely abolished GFP expression, even in the presence of the native *Sp-kirrelL* promoter, pointing to this region as the major regulatory apparatus of the gene and demonstrating that any CREs outside this region (including elements H and I) are insufficient to support transcription during embryogenesis. Consistent with plasmid reporter assays, our quantitative BAC analysis confirmed that elements C and G both make major contributions to *Sp-kirrelL* expression. Furthermore, we confirmed that the *Sp-kirrelL* native promoter is required for BAC reporter activity, also consistent with our plasmid reporter assays and with the hypothesis that the CREs are brought into physical contact with the promoter by chromatin looping during transcription. We observed that deletion of element H, which consisted of the *Sp-kirrelL* 3′-UTR, also resulted in decreased expression of the BAC reporter at 30 and 65 hpf. Although an exogenous polyadenylation site was inserted at the 3′ end of the reporter coding sequence during BAC recombineering and was therefore present in all constructs, we cannot exclude the possibility that transcription extended beyond this site and that deletion of the 3′-UTR influenced the processing or stability of the *Sp-kirrelL* transcript rather than transcription.

Elements B, E, F, and I each drove PMC-specific reporter expression in plasmid constructs that contained the *Sp-kirrelL* promoter, but deletion of these elements individually from the *Sp-kirrelL* BAC did not quantitatively affect reporter expression at the developmental stages we examined. There are several possible explanations for this. First, these CREs may have no regulatory function in vivo. According to this view, the transcriptional activity of these elements in plasmid constructs was an artifact of bringing them in close proximity to the native *Sp-kirrelL* promoter. This view is inconsistent, however, with the fact that most of these elements (B, E, and I) contain other signatures of enhancer activity. All three elements are hyperaccessible in PMCs relative to other cell types at 24 hpf as assayed by ATAC-seq, and elements E and I are also associated with eRNA signal during early development (*Figure 1*). Moreover, these elements exhibited some degree of promoter specificity in our reporter assays; that is, they were active in combination with the *Sp-kirrelL* promoter but not the *Sp-endo16*

core promoter. These findings suggest that some or all of these elements ordinarily have a regulatory function. They may modulate the precision of *Sp-kirrelL* expression during early development in subtle ways that our assays did not detect (*Lagha et al., 2012*) or they may be entirely redundant; that is, deletion of any one of these elements may result in the complete assumption of its function by other elements. This might be the case, for example, if functionally equivalent CREs ordinarily share the *Sp-kirrelL* promoter. In support of this hypothesis, many examples of functionally redundant enhancers have been described in other model systems (*Kvon et al., 2021*). Lastly, although these elements are associated with eRNA expression and cell type-specific accessibility early in embryogenesis, their primary function may be to regulate *Sp-kirrelL* expression during stages of development later than those assayed in this study.

## Coregulation of elements C and G by Alx1 and Ets1

The results of both plasmid- and BAC-based reporter assays showed that elements C and G provide crucial inputs into *Sp-kirrelL*. Detailed dissection of these key elements identified consensus Ets1- and Alx1-binding sites that were essential for activity. This finding was consistent with previous evidence that perturbation of *alx1* or *ets1* function using antisense morpholinos results in a dramatic reduction of *Sp-kirrelL* expression (*Rafiq et al., 2014*). Moreover, ChIP-seq studies have shown that Alx1 binds directly to both elements (*Khor et al., 2019*). We cannot, however, exclude the possibility that other ETS and homeodomain family members expressed in PMCs (e.g., Erg and Alx4) also bind to these sites. Interestingly, although paired-class homeodomain proteins (including Alx1-related proteins found in vertebrates) are thought to regulate transcription primarily through their binding to palindromic sites that contain inverted TAAT sequences (e.g., ATTANNNTAAT), we identified a half site (ATTA) in element C that was required for activity. This finding supports other recent work which has shown that half sites play a more prominent role in the transcriptional activity of Alx1 than was previously appreciated (*Guerrero-Santoro et al., 2021*).

Based on gene knockdown studies and the epistatic gene relationships they reveal, *Oliveri et al., 2008* proposed that several PMC effector genes are regulated through a feed-forward circuit involving Alx1 and Ets1. They showed that Ets1 positively regulates *alx1* and that both regulatory inputs are necessary to drive expression of several biomineralization-related genes. Our findings support such a model and extend it by demonstrating that the topology of this feed-forward regulation is very simple – both Alx1 and Ets1 provide direct, positive inputs into CREs associated with *Sp-kirrelL*. We identified dual, direct inputs into two different CREs, one associated with the promoter (element G) and a more distal element (element C). Evidence from other recent studies suggest that direct coregulation by Alx1 and Ets1 is a widespread mechanism for controlling PMC effector gene expression. Genome-wide analysis of Sp-Alx1 ChIP-seq peaks located near effector gene targets showed that both Alx1 and Ets1 consensus binding sites were highly enriched in these regions (*Khor et al., 2019*) and both Alx1- and Ets1-binding sites are enriched in regions of chromatin that are hyperaccessible in PMCs relative to other cell types (*Shashikant et al., 2018b*). Our analysis of *Sp-kirrelL* reveals that feed-forward regulation by Alx1 and Ets1 controls not only the expression of biomineralization-related genes but also genes that regulate PMC behavior, thereby integrating these cellular activities.

*Davidson, 1986* proposed that sea urchins, ascidians, nematodes, and several other animal groups develop by a so-called 'Type I' mechanism, a mode of development characterized by the early embryonic expression of terminal differentiation genes. A prediction of this model is that Type I embryos deploy developmental GRNs that are relatively shallow; that is, there are few regulatory layers between cell specification and cell differentiation. The *cis*-regulatory control of *Sp-kirrelL* by Alx1 and Ets1 supports this prediction; both transcription factors are activated during early embryogenesis and provide direct, positive inputs into *Sp-kirrelL*. Although mutations of other putative transcription factor-binding sites in elements C and G did not result in any noticeable effects on reporter expression in our studies, it should be noted that perdurance of GFP mRNA or protein following activation by early regulatory inputs such as Alx1 and Ets1 might have masked effects of such mutations on later stages of embryogenesis.

## Evolutionary conservation of echinoderm *kirrelL* CREs

All adult echinoderms have elaborate, calcitic endoskeletons, but larval skeletal elements are found only in echinoids, ophiuroids, and holothuroids (the latter form only a very rudimentary larval skeleton).

It is widely believed that the adult skeleton was present in the most recent common ancestor of all echinoderms and that larval skeletons arose subsequently through a developmental redeployment of the adult program (see reviews by *Cary and Hinman, 2017*; *Koga et al., 2014*; *Shashikant et al., 2018a*). It is debated, however, whether this redeployment occurred only once, with a subsequent loss of larval skeletons in asteroids, or more than once, with larval skeletons appearing independently in several groups. Our studies establish *kirrelL* as a component of the ancestral echinoderm skeletogenic GRN, which also included *alx1*, *ets1*, and *vegfr-10-Ig* (*Erkenbrack and Thompson, 2019*; *Shashikant et al., 2018a*).

There is abundant evidence that mutations in *cis*-regulatory sequences contribute to phenotypic evolution (*Rebeiz and Tsiantis, 2017*; *Wray, 2007*). At the same time, there are examples of evolutionarily conserved GRN topologies and transcription factor-binding sites, often between relatively recently diverged taxa (e.g., mice and humans) but sometimes more deeply conserved (*Rebeiz et al., 2015*). In the present study, we showed that noncoding sequences upstream of the translational start sites of *kirrelL* genes from a diverse collection of echinoderms supported PMC-specific reporter expression in sea urchin embryos. These echinoderms included a crinoid (*A. japonica*) and two sea stars (*A. planci* and *P. miniata*), taxa that diverged from echinoids 450–500 million years ago (*Paul and Smith, 1984*; *Pisani et al., 2012*). The deep evolutionary separation of these groups reveals a remarkable conservation of the *kirrelL* regulatory apparatus over this vast time period. To our knowledge, this is only the second reported case of conserved regulatory element function among deeply divergent echinoderms (*Hinman et al., 2007*). Although the amino acid sequences of KirrelL proteins are well conserved within the phylum (*Figure 6—figure supplement 2A*), the sequences of the upstream regulatory regions we identified are more divergent. Despite limited nucleotide sequence conservation, dissection of the *Pm-kirrelL* regulatory region provided evidence that in sea stars, as in sea urchins, Alx1 and Ets1 provide direct, positive inputs into *kirrelL*. Moreover, we showed that regulatory elements directly upstream of the *Pm-kirrelL* translation start site could substitute for the native *Sp-kirrelL* promoter in supporting the activity of the *S. purpuratus* C element, an effect that we hypothesize reflects a deep conservation of the binding sites and proteins that mediate CRE-promoter tethering.

The embryonic skeletogenic GRN of sea urchins has been elucidated in considerable detail, but analysis of the ancestral, adult program has this far been limited to comparative gene expression studies, as there are several technical hurdles to molecular perturbations of adult echinoderms. Because sea stars do not express *kirrelL* at embryonic stages and lack a larval skeleton, but express *kirrelL* in adult skeletogenic centers, we conclude that the function of the sea star *kirrelL* cis-regulatory system is to control the transcription of this gene in the adult. Thus, our identification of Alx1 and Ets1 inputs into the *Pm-kirrelL* regulatory region provides evidence that these inputs are required in skeletal cells of the adult sea star, consistent with the finding that both Ets1 and Alx1 are expressed selectively by these cells (*Gao and Davidson, 2008*). We cannot exclude the possibility that the regulatory interactions we detected in the context of the *S. purpuratus* embryo are vestiges of an ancient, larval skeletogenic program that has since been lost in sea stars, if indeed this was the evolutionary trajectory of larval skeletogenesis within echinoderms. This interpretation, however, would require the evolutionary conservation of the relevant regulatory DNA sequences over a vast period of time despite their complete lack of function, a scenario that seems very unlikely. We propose instead that our findings provide the first glimpse of functional gene interactions in the ancestral, adult echinoderm skeletogenic program and highlight the remarkable conservation of this program in adults and embryos. As Ets1 is expressed in the embryonic mesenchyme of modern sea stars (*Koga et al., 2010*; *McCauley et al., 2010*), our findings support the view that a major event in the co-option of the adult skeletogenic GRN into the embryo was a heterochronic shift in the expression of Alx1. This would have been sufficient to transfer a large part of the skeletogenic GRN into the embryo, as the transcription of many key effector genes, including *kirrelL*, was already directly linked to Alx1 and Ets1 expression. Direct analysis of CRE structure and function in the adult skeletogenic centers of sea stars and sea urchins will be required to more fully elucidate the architecture of the ancestral network.

## Materials and methods

### Animals

Adult *S. purpuratus* and *P. miniata* were acquired from Patrick Leahy (California Institute of Technology, USA). Adult *L. variegatus* were acquired from the Duke University Marine Laboratory (Beaufort, NC, USA) and from Pelagic Corp. (Sugarloaf Key, FL, USA). Spawning of gametes was induced by intracoelomic injection of 0.5 M KCl. *S. purpuratus* and *P. miniata* embryos were cultured in artificial seawater (ASW) at 15°C in temperature-controlled incubators while *L. variegatus* embryos were cultured at 19–24°C. Late-stage *L. variegatus* and *P. miniata* larvae were fed with *Rhodomonas lens* algae, accompanied by water changes every other day.

### Generation of *cis*-regulatory reporter constructs

Phylogenetic footprinting between echinoderm kirrelL loci was performed using GenePalette (*Smith et al., 2017*) with a sliding window size of 13–15 bp. GFP reporter constructs were generated by cloning putative CREs into the *EpGFPII* plasmid, which contains the basal promoter of *Sp-endo16* (*Cameron et al., 2004*). Putative *Sp-kirrelL* CREs were amplified from *S. purpuratus* genomic DNA using primers with restriction site overhangs (see *Figure 6—source data 1*). CREs with mutations of putative transcription factor-binding sites and putative CREs from echinoderm species were synthesized as gBlock gene fragments with flanking restriction sites by Integrated DNA Technologies (Coralville, IA, USA). Sequences of putative CREs from echinoderm species (other than sea urchins) were located directly upstream of the *kirrelL* gene translational start sites (see *Figure 6—source data 2*; *Arshinoff et al., 2022*; *Long et al., 2016*).

### BAC recombineering

*Sp-KirrelL* BAC-GFP reporter constructs were generated from a parental BAC (R3-28J10-14544) according to established recombineering protocols (*Buckley et al., 2018*). The recombineering cassettes were synthesized by Integrated DNA Technologies (Coralville, IA, USA). The cassettes contained GFP coding sequence, SV40 terminator sequence, a kanamycin resistance gene between two flippase recognition target sites and flanking homologous arms. The recombineering cassettes were transformed into EL250 cells carrying the parental BAC (pBACe3.6 vector harboring *Sp-kirrelL* and flanking genomic sequences) and recombinase genes were derepressed via heat shock. EL250 cells with recombinant BACs were selected based on kanamycin resistance. To remove the kanamycin resistance gene, expression of *flippase* (*flp*) recombinase enzyme was induced using L-(+)-arabinose and colonies with the kanamycin resistance gene removed were identified by replica plating. BACs without kanamycin resistance gene were subsequently electroporated and propagated in DH10β cells.

### Microinjection

Microinjection of reporter constructs was performed following established protocols (*Arnone et al., 2004*). Prior to injection, reporter constructs were linearized and mixed with carrier DNA that was prepared by overnight HindIII digestion of *S. purpuratus* or *L. variegatus* genomic DNA. BAC and plasmid constructs were linearized with AscI and KpnI restriction enzymes, respectively. Each 20 μl injection solution contained 100 ng linearized DNA, 500 ng carrier DNA, 0.12 M KCl, 20% glycerol, 0.1% Texas Red dextran in DNAse-free, sterile water. *S. purpuratus* embryos were cultured for 48 hpf and *L. variegatus* were cultured for 28 hpf before being mounted for live imaging. Embryos were scored to determine the total number of injected embryos (indicated by the presence of Texas Red dextran), the number of embryos showing PMC-specific GFP expression, the number of embryos showing PMC and ectopic GFP expression, and the number of embryos with only ectopic GFP expression. Microinjection of morpholinos (MOs) (Gene Tools, LLC, Philomath, OR, USA) into fertilized sea urchin eggs was performed as described (*Cheers and Ettensohn, 2004*). MO sequences (5′– 3′) were: *Sp-alx1* MO, TATTGAGTTAAGTCTCGGCACGACA; *Sp-ets1* MO, GAACAGTGCATAGACGCCAT GATTG. MOs were injected at concentrations of 3 mM (*Sp-alx1* MO) and 2 mM (*Sp-ets1* MO).

### NanoString analysis

Direct quantitative measurement of GFP and mCherry RNA transcripts and incorporated DNA was performed using the Nanostring nCounter Elements XT protocol. Briefly, a pair of target-specific

oligonucleotide pairs (Probes A and B) complementary to each target gene and transcript were synthesized by Integrated DNA Technologies (Coralville, IA, USA). Probes A and B also included short tails complementary to NanoString Reporter Tags and Universal Capture Tags, respectively. RNA targets included GFP, mCherry, and several *S. purpuratus* housekeeping genes (*foxJ1*, *hlf*, *kazL*, and *rasprp3*) that represented a range of transcript abundances and that were expressed at constant levels over the developmental time window of interest. DNA targets included GFP, mCherry, several endogenous, single-copy genes (*hypp_1164*, *hypp_1901*, *hypp_2956*, *hypp_592*, *kirrelL*), and one multicopy gene (*pmar1*). DNA probes were complementary to the noncoding DNA strand to avoid hybridization to RNA. Probe sequences are available in *Figure 5—source data 3*. For detection, we used the NanoString Elements XT Reporter Tag Set-12 and Universal Capture Tag.

Embryos injected with parental and mutant BACs were harvested at 20, 30, 50, and 65 hpf using the Qiagen AllPrep DNA/DNA micro kit. An additional on-column DNase treatment was included in the RNA recovery process to remove contaminating DNA. Genomic DNA extracted was sonicated using a Bioruptor Pico (Diagenode) for 6 min (30 s ON, 30 s OFF) at 4°C to obtain ~200 bp fragments (confirmed using an Agilent Bioanalyzer). Sonicated DNA was extracted using ethanol precipitation. GFP or mCherry RNA counts were first normalized to housekeeping transcript counts. DNA counts were normalized to single-copy gene counts to obtain number of incorporated DNA per nucleus. To obtain RNA count per incorporated DNA for each sample, normalized RNA counts were divided by normalized incorporated DNA counts (*Figure 5—source data 1* and *Figure 5—source data 2*).

## Whole-mount in situ hybridization

DNA templates for RNA probe synthesis were amplified with reverse primers that contained T3 promoter. Invitrogen MEGAscript T3 Transcription Kit was then used to amplify digoxigenin-labeled RNA from the DNA templates. WMISH was performed as previously described (*Ettensohn et al., 2007*), with minor modifications. Embryos were collected fixed at the desired stage and fixed 4% (paraformaldehyde PFA) in ASW for 1 hr at room temperature. The embryos were then washed twice in ASW and permeabilized and stored in with 100% methanol. Embryos were then rehydrated and incubated with 1 ng/μl RNA probe overnight at 55°C. The following day, the embryos were incubated in blocking buffer (1% BSA (bovine serum albumin) and 2% horse serum in PBST (phosphate-buffered saline containing 0.05% Tween-20)) and then in blocking buffer with 1:2000 α-DIG-AP antibody. Excess antibody was washed away and color reaction for alkaline phosphatase was carried out.

## Acknowledgements

We are grateful to Dr. Jennifer Guerrero-Santoro for operating the NanoString nCounter. This work was supported by grants from the National Institutes of Health (R24-OD023046) and the National Science Foundation (IOS2004952), both to C.A.E.

## Additional information

### Funding

| Funder | Grant reference number | Author |
|---|---|---|
| National Institutes of Health | R24-OD023046 | Charles A Ettensohn |
| National Science Foundation | IOS2004952 | Charles A Ettensohn |

The funders had no role in study design, data collection, and interpretation, or the decision to submit the work for publication.

### Author contributions

Jian Ming Khor, Conceptualization, Data curation, Formal analysis, Investigation, Methodology, Resources, Validation, Visualization, Writing - original draft; Charles A Ettensohn, Conceptualization, Funding acquisition, Supervision, Writing - original draft, Writing - review and editing

**Author ORCIDs**
Jian Ming Khor http://orcid.org/0000-0002-1428-6770
Charles A Ettensohn http://orcid.org/0000-0002-3625-0955

**Decision letter and Author response**
Decision letter https://doi.org/10.7554/eLife.72834.sa1
Author response https://doi.org/10.7554/eLife.72834.sa2

## Additional files

### Supplementary files
• Transparent reporting form

### Data availability
All raw numerical data used in this study are contained in the manuscript.

The following previously published datasets were used:

| Author(s) | Year | Dataset title | Dataset URL | Database and Identifier |
|---|---|---|---|---|
| Shashikant T, Ettensohn CA, Khor JM | 2018 | Global analysis of primary mesenchyme cell cis-regulatory modules by chromatin accessibility profiling | https://bmcgenomics.biomedcentral.com/articles/10.1186/s12864-018-4542-z | NCBI Gene Expression Omnibus, GSE96927 |
| Khor JM, Guerrero-Santoro J, Douglas W, Ettensohn CA | 2021 | Global patterns of enhancer activity during sea urchin embryogenesis assessed by eRNA profiling | https://genome.cshlp.org/content/early/2021/08/24/gr.275684.121.long | NCBI Gene Expression Omnibus, GSE169227 |
| Khor JM, Guerrero-Santoro J, Ettensohn CA | 2019 | Genome-wide identification of binding sites and gene targets of Alx1, a pivotal regulator of echinoderm skeletogenesis | https://journals.biologists.com/dev/article/146/16/dev180653/224197/Genome-wide-identification-of-binding-sites-and | NCBI Gene Expression Omnibus, GSE131370 |

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
