## [Editor Report]

In this manuscript, Khor et al. examine the transcriptional regulation of kirrelL, a gene whose protein product is required for cell-cell fusion during the morphogenesis of the sea urchin larval skeleton. They establish a putative direct link between a developmental gene regulatory network driving cell fate commitment and an effector protein enabling a key behavior of the specified cell type, thereby strengthening the explanatory power of a well-established GRN model. It places a key morphoregulatory gene, kirrelL, into the extensively studied gene regulatory network of sea urchins and reveals deep evolutionary conservation of regulatory element function. This study should be of broad, general interest for developmental biologists.

---

## [Decision Letter]

**Decision letter after peer review:**

Thank you for submitting your article "Architecture and evolution of the *cis*-regulatory system of the echinoderm *kirrelL* gene" for consideration by *eLife*. Your article has been reviewed by 3 peer reviewers, and the evaluation has been overseen by a Reviewing Editor and Kathryn Cheah as the Senior Editor. The reviewers have opted to remain anonymous.

The reviewers have discussed their reviews with one another, and the Reviewing Editor has drafted this to help you prepare a revised submission. The reviewers are in general agreement that the manuscript warrants publication in *eLife*. However, the reviewers have also requested some additional experiments and edits to improve the study and the manuscript. We suggest the authors address all of the issues raised by the individual reviewers, giving special attention to the essential revisions listed below.

Essential revisions:

1) The reviewers feel that is it necessary to validate RNA-seq results with qPCR or in situ hybridization when possible, and so agree that it would be appropriate for the authors to knock down Alx1 in S. purpuratus embryos both to confirm that kirrelL expression decreases and to show that the activity of the P. miniata CRE is affected. Likewise, the reviewers feel that it will be important for the authors to show that Alx1 knockdown affects the activity of elements C and G to rule out the possibility that other transcription factors' binding to these sites underlies their importance.

2) The reviewers agree that the authors' data suggests that the Sp-kirrelL promoter is necessary for the BAC construct to be expressed. However, they believe that the authors need to revise the statement in line 252, where it is stated that endo16prm+Sp-kirrelLprm drives ectopic expression, and that this suggests the kirrelL promoter is a strong, ubiquitous promoter. The reviewers think the conclusion the authors draw from this reporter assay contradicts their BAC experiment showing that the promoter alone (without the other elements) cannot drive expression.

3) The reviewers also agree that it is important for the authors to analyze sequence conservation of the studied species' kirrelL cis-regulatory elements to support the claim of conserved sequence and function.

*Reviewer #1:*

In this manuscript, Khor et al. examine the transcriptional regulation of kirrelL, a gene whose protein product is required for cell-cell fusion during the morphogenesis of the sea urchin larval skeleton. They identify cis-regulatory elements of Sp-kirrelL that contain putative binding sites for Alx1 and Ets1, two transcription factors that specify the cellular precursors of the larval skeleton, and show that mutating these sites abrogates one element's ability to drive reporter expression and alters the spatial pattern of the other. This putative direct link between a developmental gene regulatory network driving cell fate commitment and an effector protein enabling a key behavior of the specified cell type would strengthen the explanatory power of a well-established GRN model, making this study of interest for developmental biologists broadly.

The authors' selection of candidate cis-regulatory regions is well-grounded in functional genomic data showing that several of these regions possess signatures of active cis-regulatory elements. A reporter construct containing all of the chosen regions (A-G) is expressed specifically in primary mesenchyme cells in the majority of expressing, injected embryos, consistent with the PMC-specific expression pattern of Sp-kirrelL, confirming that this genomic region very likely contains enhancers of Sp-kirrelL. A thorough dissection of A-G identifies two regions individually able to drive PMC-specific expression when placed in front of a basal promoter, albeit at low frequency, leading the authors to investigate the trans-regulation of these elements. Elements C's loss of activity upon mutation of a single Alx site or a single Ets site is an exciting finding. It is interesting as well that mutating Alx or Ets sites within G causes it to drive ectopic activity. If, as these data suggest, Alx1 and Ets1 directly regulate the expression of Sp-kirrelL, this would be one of few sea urchin studies that have shown a direct and functional cis-regulatory link from a fate-specifying gene to an effector gene expressed in the differentiated cell (others are Amore and Davidson Dev. Biol. 2006 and Calestani and Rogers Dev. Biol. 2010, though the latter relies on mutation of putative binding sites only). Additionally, the high resolution at which the function of kirrelL is understood-we know the specific cell behavior it is needed for-is strong justification for studying the regulation of this gene, and therefore these findings on kirrelL's regulation are an important contribution to understanding of PMC morphogenesis. However, to confirm that Alx1 and Ets1 directly regulate Sp-kirrelL, it is necessary to show that knockdown of either factor alters Sp-kirrelL's expression, since the putative binding sites the authors identify could be bound by other transcription factors. Characterizing the effects of Alx1 and Ets1 knockdown on cis-regulatory element reporter activity could also further support the authors' conclusions.

The analysis of individual cis-regulatory elements' necessity for Sp-kirrelL expression using bacterial artificial chromosome deletions is a well-justified approach to understanding these elements' functions. Elements C, G and H each contribute to transcription in a BAC context and therefore likely contribute non-redundantly to endogenous Sp-kirrelL expression, though statistical analysis of the Nanostring data is needed to confirm the significance of these effects.

Aiming to test whether the Alx/Ets-kirrelL cis-regulatory link proposed for S. purpuratus is conserved in skeletal development of other echinoderm species, the authors characterize the expression of kirrelL in the sea star P. miniata and perform a heterologous reporter experiment showing that mutating putative Alx and Ets sites in a region upstream of Pm-kirrelL abolishes the region's ability to drive PMC-specific reporter expression. This result and the fact that Pm-kirrelL, Alx1 and Ets1 are expressed in PMCs in P. miniata are consistent with the authors' claim that Alx1 and Ets1 regulate kirrelL in P. miniata. However, given that the trans environments of S. purpuratus embryos and P. miniata larvae likely differ, the evidence shown so far is not compelling support for this claim. Showing that knockdown of Alx1 or Ets1 in S. purpuratus embryos alters the activity of the Pm-kirrelL CRE is necessary to support the assertion that these proteins regulate this element in its native species and developmental context.

Comments for the authors:

Evidence for direct regulation by Alx1 and Ets1:

As raised in the public review, experiments testing how perturbing Alx1 and Ets1 expression affects Sp-kirrelL expression are needed to confirm that (1) binding of Alx1 and Ets1 (and not another factor) to the altered CRE sites underlies these sites' importance and (2) that this interaction consequential for the expression of Sp-kirrelL. The authors raise the former issue in lines 552-554. Though Rafiq et al. (2014) provide evidence for the latter in RNA-seq data, this should be validated with in situ hybridization, which will also allow reveal any changes in spatial expression of kirrelL (e.g. ectopic expression, which seems plausible given that mutating certain sites leads to ectopic reporter activity). An analogous experiment with the Pm-kirrelL CRE is necessary to support the claim that Alx1 and Ets1 regulate Pm-kirrelL.

Reporter assay data:

– Why is there so much variation in the number of embryos injected per construct?

– Does element C consistently fail to drive reporter expression in the vegetal-most PMCs, as in the image in Figure 2C? If so, this should be noted.

– Please state in the caption of Figure 2 the exact percent of GFP+ embryos that needed to have PMC-specific expression for a construct to be classified as strong or weak PMC expression (I assume 50%). Furthermore, by the criteria for being classified as "ectopic expression" in Figure 4, C.ChIP.Alx1palindrome should fall into this category. These classification criteria should be applied consistently to all figures.

– To support the conclusion that mutating putative Alx1 or Ets1 binding sites alters the spatial activity of the Sp-kirrelL promoter and of PmG (lines 254-255 and 435-7 state that these elements receive positive inputs from Alx1 and Ets1), the fraction of expressing embryos that display ectopic activity should be compared statistically between the wild-type and mutant construct for the Sp-kirrelL promoter and for PmG.

– Please state sample sizes for the experiment in which S. purpuratus CRE constructs were injected into L. variegatus embryos (of which images are shown in Figure S2B).

Nanostring data:

– Supplemental Table S1 appears to be missing expression pattern data for all but one of the BACs. These data are necessary to support the statements made in lines 336-343 on the spatial activity patterns of the deletion constructs.

– Statements comparing expression levels driven by the wild-type BAC and a particular deletion BAC (the authors make such statements for constructs ∆C.GFP.BAC, ∆G.kirrelLprm.GFP.BAC, and ∆H.GFP.BAC in lines 348-355) should be supported with statistical analysis of the nanostring data presented in figure 5C. This is especially necessary given that the normalized RNA counts for some replicates and time points are quite similar between the compared constructs (e.g. ∆C.GFP.BAC and the wild-type BAC at 50 hpf).

Suggestion for presentation of data:

The figures in their current form succeed in condensing the results of many reporter assays into a visually digestible form. However, conveying the results of each assay using a single parameter (strong, weak, or no PMC expression, or ectopic expression) whose value is determined using thresholds for percent of GFP+ embryos and percent of GFP+ embryos with PMC-specific expression hides interesting differences in constructs’ activity. For example, though C.ChIP, C.ChIP.Alx1halfsite1 and C.ChIP.Alx1palindrome are all categorized as “strong PMC expression”, the mutant constructs drive reporter activity in PMCs only less frequently and in ectopic locations more frequently than the wild-type construct, suggesting that these sites may be relevant for proper Sp-kirrelL expression.

I would suggest making bar plots depicting the spatial pattern category data in Supplemental Table S1 and placing these in the supplemental figures, or in the main figures where the data are most relevant or exciting (e.g. site mutations leading to ectopic activity). I would also consider graphically depicting for each construct the proportion of embryos showing expression. These figures would aid readers in making comparisons within these data, which may be of interest to those studying cis-regulation broadly.

Issue with interpretation of data relating to Sp-kirrelL promoter:

The claim in line 251, that the Sp-kirrelL promoter is a strong, ubiquitous promoter, is contradicted by the result in Figure 5a and lines 326-328 that a BAC containing the promoter only does not drive any expression. The latter result suggests that the Sp-kirrelL promoter lacks the ability to drive transcription without additional enhancer elements, and that the strong activity of the Sp-kirrelL promoter + endo16 promoter construct is due to synergistic activity of the Sp-kirrelL and endo16 promoters.

Issue pertaining to claim of enhancer-promoter-specific interactions:

The statement "several CREs are capable of interacting specifically with the native Sp-kirrelL promoter” (274-276) should be rephrased and clarified. I would not use the word “interact” to describe an observation about the output of a combination of two CREs because this word suggests physical CRE-promoter looping, for which no direct evidence is presented. Wording like “super-additive/synergistic activity” would be more clear.

Furthermore, in claiming that pairs of elements interact specifically/display super-additive activity, the authors rightfully state that, e.g., B+sp-kirrelLprm+endo16prm displays activity while B+endo16prm does not drive any reporter expression. However, given that sp-kirrelLprm+endo16prm itself drives expression, the activity of B+sp-kirrelLprm+endo16prm must be greater than the sum of the activities of B+endo16prm and sp-kirrelLprm+endo16prm in order to state that B and sp-kirrelLprm interact super-additively. In other words, the activity of kirrelLprm+endo16prm must be taken into account.

Considering this activity, it does appear that B, C, E, F, H and I do act super-additively with Sp-kirrelLprm, considering at least one of the metrics of % of embryos expressing and % with PMC-specific expression (e.g. C+endo16prm drives GFP expression in 8.6% of embryos, Sp-kirrelLprm+endo16prm drives expression in 19%, and C+Sp-kirrelLprm+endo16prm drives expression in 37.8%, which is greater than 19+8.6). A+Sp-kirrelLprm+endo16prm and D+Sp-kirrelLprm+endo16prm, on the other hand, drive GFP expression both with less frequency and less PMC specificity than Sp-kirrelLprm+endo16prm, so I do not think it can be said that these elements interact specifically with kirrelLprm, and it seems that they do not have enhancer activity and may not be functional elements at all. If this is why BAC deletion constructs were not made for A and D, this should be explicitly stated.

It is interesting that several X+Sp-kirrelLprm+endo16prm combinations are more PMC-specific than Sp-kirrelLprm+endo16prm alone. This suggests that some of these elements may not only be enhancers, but also have silencer activity necessary to drive kirrelL expression in the proper spatial pattern. It is an interesting finding, in line with Gisselbrecht et al. Mol. Cell. (2020)'s finding that many silencers are enhancers in alternate tissue contexts, that should be highlighted.

Finally, the interpretation in lines 269-271 that "the presence of the native Sp-kirrelL promoter mitigated the need for the C.ChIP element within element C to be adjacent to the promoter for strong PMC-specific GFP expression" (and restated in 274-276) is not strongly supported. Placing a spacer sequence with no enhancer or silencer activity in between C.ChIP and the promoter, and showing that this decreases activity as compared to when C.ChIP is adjacent to the promoter, would support this. However, the current data leave open the possibility that the part of element C downstream of C.ChIP contains sites for repressive factors, especially given that there is such a large difference in the activity of C.DNase and C.ATAC even though the former is only ~150 bp longer.

While these may seem like small points, being precise with the interpretation of the wealth of reporter assay and BAC data in this study will elevate the study's contribution to the field of cis-regulation.

kirrelL gene/protein tree:

Ettensohn and Dey Dev. Biol. (2017) show a protein tree in which Sp-kirrelL, Lv-kirrelL and Pm-kirrelL cluster together within a set of transmembrane Ig-domain proteins, but I have not found such an analysis for the sequences of kirrelL proteins in the other species whose kirrelL cis-regulatory regions are tested for activity in this manuscript. This is necessary to confirm the identity of these kirrelL genes. If such an analysis has been published, that paper should be cited.

Description of cross-species experiments:

The sequences or coordinates of the putative kirrelL CREs from multiple species tested in reporter assays should be provided. When describing the cross-species CRE reporter experiments (lines 376-381), please explicitly state the species for which the expression pattern of kirrelL is unknown, as this is relevant to the interpretation of these experiments. Also, the choice to examine the expression of kirrelL in the adult rudiment of L. variegatus rather than that of S. purpuratus needs to be explained.

Finally, when presenting the results of the promoter swap experiment, I would explicitly state that PmG1 and PmG2 each confer a roughly similar increase in expression frequency and specificity to C+endo16prm as Sp-kirrelLprm does to more clearly convey that PmG1 and G2 can substitute for Sp-kirrelLprm in a meaningful way.

*Reviewer #2:*

The embryonic gene regulatory network (GRN) of sea urchins has been studied in considerable detail, providing numerous insights into how GRNs contribute to the development and evolution of phenotype. This study is significant because it places kirrelL, an important morphoeffector gene, into the GRN, thus providing an important link between early regulatory interactions that pattern the embryo and specify cell fate and later interactions that activate the genes that carry out morphogenesis. Through detailed experimental analyses, the authors identified the cis-regulatory elements that control the precise spatial and temporal pattern of kirrelL transcription and two transcription factors that act as positive inputs. A strength of the study is the combination of plasmid-based and BAC-based expression assays that dissect in detail the contribution of individual and combinations of regulatory elements, as well as targeted deletion of predicted transcription factor binding sites. Minor weaknesses are that most of these experiments used a heterologous basal promoter rather than the kirrelL basal promoter and that they were not designed to detect repressive interactions. Despite these minor concerns, the results identify the principal cis-regulatory elements and trans inputs that control kirrelL expression in the sea urchin embryo. A second set of experiments tested the ability of the 5' flanking region of the kirrelL gene from other echinoderm species to drive reporter gene expression. The species tested include members of distantly related groups and ones whose larvae do not produce a skeleton. The results show that the 5' upstream region contains regulatory elements that activate spatially and temporally correct transcription in the sea urchin embryo. Although regulatory element that have been conserved in function over comparable time scales have been described in other groups of animals, this seems to be the first well-documented example from echinoderms. The authors describe this as a case of "striking conservation of sequence and function" although it is not clear from the evidence presented that sequence conservation is actually involved. The authors show evidence of limited sequence conservation in noncoding regions around kirrelL between two sea urchins and between two sea stars, but no evidence for sequence conservation among the major groups of echinoderms. Even in the more closely related species there is no evidence that the small patches of similar sequences are actually the basis for conserved regulatory function. An interesting finding is that even species whose larvae lack a skeleton contain a 5' flanking region that can drive spatially and temporally accurate transcription in the sea urchin embryo. This finding led to the discovery of an enhancer that directs expression in the adult skeleton within the 5' flanking region of sea stars. Together, these results hint that some of the transcription factors that activate kirrelL transcription in embryos also perform that function during skeletogenesis in adult echinoderms.

Overall, this is a beautifully conducted study. The results are presented very clearly in both text and figures. Some specific questions and concerns follow below (numbers refer to line numbers in the manuscript):

178. What is the justification for using the core promoter of a different gene for these experiments? Given that some enhancers show selectivity for nearby core promoters (at least in other systems), this seems like an odd choice.

Figure 2B. I couldn't find any information about the number of replicates for these experiments, or any of the subsequent reporter assays presented in subsequent figures. A detailed tabulation for every separate experiment is not necessary, but a general statement about the number of replicates in a typical experiment would be very reassuring.

196-198. The explicit criteria for defining "strong" and "weak" expression are helpful. That said, it wasn't immediately obvious from Figure 2C how these differ when looking at the images for ABC and C (weak) versus the other constructs (strong). The weak constructs look a bit out of focus but that could simply be the weaker signal. How consistent are these differences among embryos and from replicate to replicate?

212-214. Could this result also be explained by the presence of binding sites for repressors in ABC that BC.ATAC lacks?

249-250. What was the basis for considering this region to be the core promoter?

251-252. Are these features (strong, ubiquitous) true of the core promoter that was used for the experiments? If not, the earlier concern about choice of core promoter for the experiments is even more acute.

265-268. This is an interesting result. What does it imply mechanistically?

342. Why so much redundancy? This is touched on only briefly in the Discussion but seems like an important result.

352-354. Is there any indication here (or in the previous results) of ectopic expression?

More generally: The experiments do not seem geared to detect possible repressive functions for any of the regulatory elements. Are there reasons for thinking that repressive functions would not be needed? What keeps kirrelL from being transcribed outside of the single cell type where it is expressed?

Figure 5: modifying the key in the lower left as follows might help make it easier to interpret panel C: GFP (deletion) mCherry (intact)

376: Mention the extent of region tested here so that readers don't need to consult other parts of the paper to understand the basic experimental design. If not exactly the same region in all of the species, mention that, too.

383-384. The results support function being highly conserved. What about sequence? It looks like there is some patchy short sequence conservation between two sea urchins (Figure 1C) and two sea stars (Figure 7A). What about between these groups?

In the literature "conserved regulatory element" more commonly applies to sequence than function. To avoid confusion, qualify "conserved" throughout the manuscript to clarify whether sequence or function is being discussed.

447-448. Is there evidence for sequence conservation (and see above)?

622-623. Is this sentence intended to mean that specific binding sites are conserved? Or that binding sites for the same transcription factors are present? This is an important distinction. There is evidence from other systems that individual binding sites can turn over while conserving regulatory element function. That seems much more plausible in this case than super-strong conservation of tiny patches of sequence.

There are examples from other groups (especially vertebrates) where a regulatory element shows conserved function among highly divergent species. It would be helpful to mention this in the Discussion to provide some context. Is this the first reported case of conserved regulatory element function among deeply divergent echinoderms? If so, this is worth mentioning explicitly, again for context.

*Reviewer #3:*

In an earlier paper this group showed that KirrelL is a protein necessary for syncytial fusion of skeletal cells in the sea urchin larva. Knowledge of the gene regulatory network driving expression of KirrelL showed that two transcription factors, Alx1 and Ets1 are drivers of KirrelL expression. The analysis in this paper accomplishes two goals: they uncover much of the cis-regulatory apparatus that drives KirrelL expression exclusively in the skeletogenic cells of the sea urchin larva, and they also demonstrate an unusually long period of relative conservation of the enhancer and basal promoter driving KirrelL expression.

The paper is very clearly written and illustrated. The illustrations are quite effective in showing the outcome of the experiments along the way, and each experiment is accompanied by a fluorescent read-out in the larva to show the specificity of expression of a construct. There are several surprises in the paper. The one that was most unusual to this reviewer was the deep conservation. They obtained cis-regulatory sequences from KirrelL genes in other classes of Echinoderms, including those that have no larval skeleton. These sequences were used to build constructs upstream of GFP. The constructs were injected into sea urchin eggs (S. purpuratus), and even those that have no larval skeleton, like sea stars, have a cis regulatory region that drives gene expression in skeletogenic cells in S. purpuratus, indicating a conservation of more than 500 million years. They go on to show that the sea star expresses KirrelL in the adult skeleton, apparently using the same cis-regulatory region. The paper will be of interest to several communities. It will be of strong interest to the echinoderm development community. It will also be of interest to those interested in evolution of cis-regulatory regions and their contribution to evolutionary change.

Finally, the fact that the gene is expressed in larvae that have skeletons and the same cis regulatory region drives the expression of the gene in adults of species that have no larval skeletons is a most interesting observation.

Comments for the authors:

The paper is really well written and logically presented. If it were simply a cis regulatory story on a random gene I would have recommended a lower level journal. However, here the impressive conservation of the cis regulatory region elevates the paper quite significantly. Still, they don't show anything of the cis-regulatory region sequences, and these would help improve the paper – is the conservation actually small islands perhaps the Alx1 and Ets binding regions? Is it the basal promoter primarily? Or is the entire region fairly well conserved? The answer to those questions will be helped with the sequences. I point out that illustrating the protein sequences across the phylum is OK but is not really the focus of this paper. The other point that I think needs better quantification is the call that anything above 15% expression is considered "strong". I realize that these kind of reporter assays sometimes have a low percentage read-out, but the question is whether some elements are stronger than others. I ask simply the question about the strongest of the so-called strong vs the weakest of the so-called strong.

There are several components of the paper that would benefit from some revision to help me understand some of the interpretations.

1. In Figure 1 showing ATAC-seq, DNAse hypersensitivity, ChIP seq data how did you call the peaks? Some, especially G, the promoter, are obvious but others are less so, and still others look similar to peaks you called as enhancers.

2. You indicate that in scoring expression that greater than 15% of the embryos expressing a construct is strong expression. I understand some of the many reasons why 85% might not express. But, I would also like to know if some are stronger than others. For example is ABCDEFG stronger than DEFG and is that stronger than G? Also in Figure 2 I notice that C is indicated as weaker. The embryo illustrated has fewer fluorescent PMCs than other embryos in this panel with stronger promoters. Is it possible that element C somehow operates in a restricted number of cell bodies rather than all of them? After all, you show in other papers that there are mRNAs that are expressed in a restricted subset of PMCs even within the syncytium. Along the same line, although you indicate that Ets1 sites 2 and 3, when mutated, still allow for expression. Your images of those show stronger expression of GFP in the ventrolateral clusters than the spread of expression when all three Ets sites are control. Is that, or could that be meaningful?

3. The data in Figure 5 I find to be most valuable to the paper. That includes the nanostring data to indicate contributions relative to an mCherry control construct which is most informative.

4. The data on other species is most interesting. I see you have an alignment of the KirrelL proteins of the several species but that isn't really the story. You don't have anything to indicate the relative alignment of the same cis regulatory regions. The real question is how are those related? Are their islands of high conservation as seen earlier by Cameron and Davidson? Are there indels outside the areas that must be conserved enough to drive expression in S. purpuratus? It seems to me that these are the sequences of interest. Yes, there is demonstrated similarity in the KirrelL protein sequence but this paper is all about the cis regulation.

---

## [Author Response]

Essential revisions:1) The reviewers feel that is it necessary to validate RNA-seq results with qPCR or in situ hybridization when possible, and so agree that it would be appropriate for the authors to knock down Alx1 in S. purpuratus embryos both to confirm that kirrelL expression decreases and to show that the activity of the P. miniata CRE is affected. Likewise, the reviewers feel that it will be important for the authors to show that Alx1 knockdown affects the activity of elements C and G to rule out the possibility that other transcription factors' binding to these sites underlies their importance.

As requested, we have carried out additional studies examining the effects of *Sp-alx1* knockdown on the activity of the *P. miniata* regulatory region and the *S. purpuratus* C and G elements. These data are shown in new Figure 7—figure supplement 4. The results of these studies confirm that knockdown of Alx1 or Ets1 expression substantially suppresses the activity of all four constructs. (Line 378-382)

With respect to the effect of Alx1 on *kirrelL* expression, the key observations already published are: (1) both *kirrelL* and *alx1* are expressed only in PMCs and (2) RNAseq data (from S. purpuratus) show that *kirrelL* expression declines dramatically (to <2% of control levels) in Alx1 morphants (Rafiq et al., 2014). These RNAseq data show conclusively that Alx1 is a positive regulator of *kirrelL* expression, at least in the species we have used here (*S. purpuratus)*. Nevertheless, we have carried out additional WMISH analysis of both Alx1 and Ets1 morphants and confirmed that *Sp-kirrelL* expression declines to undetectable levels in these morphants. These results are shown in new Figure 3—figure supplement 2. (Line 204-208)

2) The reviewers agree that the authors' data suggests that the Sp-kirrelL promoter is necessary for the BAC construct to be expressed. However, they believe that the authors need to revise the statement in line 252, where it is stated that endo16prm+Sp-kirrelLprm drives ectopic expression, and that this suggests the kirrelL promoter is a strong, ubiquitous promoter. The reviewers think the conclusion the authors draw from this reporter assay contradicts their BAC experiment showing that the promoter alone (without the other elements) cannot drive expression.

We thank the reviewers for detecting this discrepancy. Based on their comments, we went back and re-tested the BAC construct that lacks elements A-G except for the 301 bp promoter region (∆CRE.kirrelLprm.GFP.BAC) and confirmed that it is not expressed at significant levels, demonstrating that the promoter alone is relatively inactive. In the context of the *EpGFPII* plasmid, however, the same 310 bp fragment drives significant ectopic expression. In agreement with Reviewer 1, we conclude that in the plasmid construct there is some unexplained and abnormal synergy between the two promoters. We have modified the Results to state: “When tested in the *EpGFPII* plasmid, we found that a 301 bp region surrounding the transcriptional start site, a region we considered to include the *Sp-kirrelL* core promoter, drove ectopic GFP expression. As shown below, however, the same element failed to drive significant reporter expression in a BAC construct, indicating that the activity of the 310 bp element in EpGFPII was the result of abnormal synergy between the *Sp-kirrelL* and *Sp-endo16* promoters.” (Line 218-224)

3) The reviewers also agree that it is important for the authors to analyze sequence conservation of the studied species' kirrelL cis-regulatory elements to support the claim of conserved sequence and function.

We thank the reviewers for requesting this additional analysis. In the original submission, we included comparisons of the C and G modules of *S. purpuratus* and *L. variegatus* (these sequence comparisons are now found in Figure 3—figure supplement 1 and Figure 4—figure supplement 2). In the revised manuscript, we have added a comparison of the *P. miniata* and *S. purpuratus* regulatory regions (see new Figure 7—figure supplement 2) and short statements in the Methods and Results sections based on this analysis. The essential finding is that these sequences are highly divergent, which is perhaps not surprising given the vast evolutionary distance between the two species (>450 million years). Consensus Alx1 and Ets1 binding sites are present in both, but their number and spacing are not conserved and there are no large blocks of conserved sequence in the two regions. In contrast, the regulatory regions of two sea stars, *P. miniata* and *A. plancii*, are much more highly conserved, and we have added a detailed comparison of these two sequences in Figure 7—figure supplement 2.

Reviewer #1:[…] Evidence for direct regulation by Alx1 and Ets1:As raised in the public review, experiments testing how perturbing Alx1 and Ets1 expression affects Sp-kirrelL expression are needed to confirm that (1) binding of Alx1 and Ets1 (and not another factor) to the altered CRE sites underlies these sites' importance and (2) that this interaction consequential for the expression of Sp-kirrelL. The authors raise the former issue in lines 552-554. Though Rafiq et al. (2014) provide evidence for the latter in RNA-seq data, this should be validated with in situ hybridization, which will also allow reveal any changes in spatial expression of kirrelL (e.g. ectopic expression, which seems plausible given that mutating certain sites leads to ectopic reporter activity). An analogous experiment with the Pm-kirrelL CRE is necessary to support the claim that Alx1 and Ets1 regulate Pm-kirrelL.

Please see our response to the Editor’s letter (Point 1). We have carried out additional studies examining the effects of *Sp-alx1* knockdown on the activity of the *P. miniata* regulatory region and *S. purpuratus* C and G elements, and on *Sp-kirrelL* expression.

Reporter assay data:– Why is there so much variation in the number of embryos injected per construct?

For most constructs, we scored 100 to 200 injected embryos. In some cases, however, we performed additional trials if expression was weak, and we needed to inject more embryos in order to get a reasonable number that expressed detectable levels of GFP and could be scored for spatial expression. Also, in some cases, reporter constructs were used as internal controls in later experiments to allow direct comparisons with other constructs. In such cases, data were pooled from many replicates and the number of injected embryos is especially large.

– Does element C consistently fail to drive reporter expression in the vegetal-most PMCs, as in the image in Figure 2C? If so, this should be noted.

As we note below in response to Reviewer 3, our microscopic analysis of transgenic embryos was carried out at 48 hpf, which is soon after PMC fusion is complete. Because GFP protein diffuses rapidly throughout the PMC syncytium, the entire PMC network is labeled in the embryos shown in Figures 2, 6, Figure 3—figure supplement 1, Figure 4—figure supplement 1, 2, 3, Figure 5—figure supplement 1, Figure 7—figure supplement 1, 4, despite the mosaic incorporation of transgenes in sea urchin embryos. In the case of the embryo injected with the C construct shown in Figure 2, this embryo has an unusually small number of PMCs and the dorsal part of the syncytial ring (located at the bottom of the embryo as shown) is missing. All the PMC cell bodies are labeled with GFP, however, as one can see if the lower (DIC) image is compared to the image showing GFP fluorescence.

– Please state in the caption of Figure 2 the exact percent of GFP+ embryos that needed to have PMC-specific expression for a construct to be classified as strong or weak PMC expression (I assume 50%). Furthermore, by the criteria for being classified as "ectopic expression" in Figure 4, C.ChIP.Alx1palindrome should fall into this category. These classification criteria should be applied consistently to all figures.

To score embryos, we first identified every injected embryo based on the fluorescence of the dextran that was co-injected with the constructs. We scored as “expressing” every embryo that had any cells with detectable GFP (or mCherry) fluorescence. Each expressing embryo was then classified into one of 3 bins, depending on whether reporter expression was confined entirely to the PMC syncytium (PMC only), entirely to other cells (ectopic only), or a combination of the two (PMC + ectopic). A comprehensive table showing the data from all constructs is included as Figure 2 – source data 1.

For Figure 2 and similar figures, to be shown as “strong PMC expression”, we required (1) that the “PMC only” class be the largest of the 3 expression classes (i.e., we required that >1/3 of GFP-expressing embryos exhibit expression only in PMCs) and (2) that this expression class represented >15% of all injected embryos. We felt that the 1/3 cutoff was sufficiently stringent because typically many other embryos exhibited expression in the PMC syncytium but also had 1 or 2 ectopic cells labeled, and so fell into the “PMC + ectopic” class. To be classified as “weak PMC expression” we also required that the “PMC only” class be the largest of the 3 expression classes, but in this case the class represented <15% of all injected embryos, reflecting lower overall levels of expression. We have modified the legend to Figure 2 to make this scoring scheme clearer.

– To support the conclusion that mutating putative Alx1 or Ets1 binding sites alters the spatial activity of the Sp-kirrelL promoter and of PmG (lines 254-255 and 435-7 state that these elements receive positive inputs from Alx1 and Ets1), the fraction of expressing embryos that display ectopic activity should be compared statistically between the wild-type and mutant construct for the Sp-kirrelL promoter and for PmG.

Figure 2 – source data 1 shows that most embryos injected with the parental G.ATAC construct exhibit PMC–specific expression, while most embryos injected with G.ATAC (Axl1 sites mutated) or G.ATAC (Ets1 sites mutated) exhibit ectopic expression (i.e., only ectopic or PMC + ectopic). For each mutant construct, the difference in the distribution of embryos in these in these two phenotypic classes as compared to the parental construct is highly significant as by a chi-square test (P<0.001). The differences between the parental PmG construct and each of the two mutant constructs (all Alx1 sites mutated or all Ets1 sites mutated) are even more dramatic and are also highly significant by a chi-square test (p<0.001). We have added relevant statements to the text. (Line 228-231)

– Please state sample sizes for the experiment in which S. purpuratus CRE constructs were injected into L. variegatus embryos (of which images are shown in Figure S2B).

This information is now shown in Figure 2 – source data 1.

Nanostring data:– Supplemental Table S1 appears to be missing expression pattern data for all but one of the BACs. These data are necessary to support the statements made in lines 336-343 on the spatial activity patterns of the deletion constructs.

This information is now shown in Figure 2 – source data 1.

– Statements comparing expression levels driven by the wild-type BAC and a particular deletion BAC (the authors make such statements for constructs ∆C.GFP.BAC, ∆G.kirrelLprm.GFP.BAC, and ∆H.GFP.BAC in lines 348-355) should be supported with statistical analysis of the nanostring data presented in figure 5C. This is especially necessary given that the normalized RNA counts for some replicates and time points are quite similar between the compared constructs (e.g. ∆C.GFP.BAC and the wild-type BAC at 50 hpf).

The Nanostring experiments were laborious and, unfortunately, we only have 2 biological replicates of the time-course data for each of the eight BAC constructs tested. This means that our statistical power is low and only in the most extreme cases (like ∆G.GFP.BAC) does a simple test like a t-test support with high confidence a difference in the mean expression level between the parental and mutant constructs at any individual time point. For the two most important constructs that we emphasize in the text (∆C.GFP.BAC and ∆G.kirrelLprm.GFP.BAC), we think the overall trend is very convincing, since the expression level of the mutant construct is invariably lower than the co-injected parental construct, at every time point and in both replicates. It can be seen, however, that for both ∆C.GFP.BAC and ∆G.kirrelLprm.GFP.BAC, one trial consistently showed higher overall expression (i.e., of both the parental and mutant constructs) than the other, perhaps due to the random nature of the BAC integration, variation between egg batches, differences in injection volume, or other factors. This creates enough difference in the expression values between the two replicates that statistical analyses we’ve explored don’t support differences between the mutant and parental BACs at high confidence levels. For example, area-under-the-curve comparisons between mutant and parental constructs yields p-values of 0.08 and 0.12 for the ∆G.kirrelLprm.GFP.BAC and ∆C.GFP.BAC, respectively, so not at the usual 0.05 threshold (but not too far off). We can include these p-values, if the reviewers feel they are useful.

Suggestion for presentation of data:The figures in their current form succeed in condensing the results of many reporter assays into a visually digestible form. However, conveying the results of each assay using a single parameter (strong, weak, or no PMC expression, or ectopic expression) whose value is determined using thresholds for percent of GFP+ embryos and percent of GFP+ embryos with PMC-specific expression hides interesting differences in constructs' activity. For example, though C.ChIP, C.ChIP.Alx1halfsite1 and C.ChIP.Alx1palindrome are all categorized as "strong PMC expression", the mutant constructs drive reporter activity in PMCs only less frequently and in ectopic locations more frequently than the wild-type construct, suggesting that these sites may be relevant for proper Sp-kirrelL expression.I would suggest making bar plots depicting the spatial pattern category data in Supplemental Table S1 and placing these in the supplemental figures, or in the main figures where the data are most relevant or exciting (e.g. site mutations leading to ectopic activity). I would also consider graphically depicting for each construct the proportion of embryos showing expression. These figures would aid readers in making comparisons within these data, which may be of interest to those studying cis-regulation broadly.

We thank the reviewer for this valuable suggestion, and we have modified the figures as recommended. Bar plots of the most important data are now found in the main figures and the other bar plots are in the relevant supplemental figures. The proportion of embryos showing expression is also indicated in the bar plots. All the raw data are found in Figure 2 – source data 1.

Issue with interpretation of data relating to Sp-kirrelL promoter:The claim in line 251, that the Sp-kirrelL promoter is a strong, ubiquitous promoter, is contradicted by the result in Figure 5a and lines 326-328 that a BAC containing the promoter only does not drive any expression. The latter result suggests that the Sp-kirrelL promoter lacks the ability to drive transcription without additional enhancer elements, and that the strong activity of the Sp-kirrelL promoter + endo16 promoter construct is due to synergistic activity of the Sp-kirrelL and endo16 promoters.

We agree and thank the reviewer for detecting this. Please see our response to Point #2 of the Essential Revisions outlined in the Editor’s letter.

Issue pertaining to claim of enhancer-promoter-specific interactions:The statement "several CREs are capable of interacting specifically with the native Sp-kirrelL promoter" (274-276) should be rephrased and clarified. I would not use the word "interact" to describe an observation about the output of a combination of two CREs because this word suggests physical CRE-promoter looping, for which no direct evidence is presented. Wording like "super-additive/synergistic activity" would be more clear.

We have changed “interacting” to “functioning in concert”. (Line 259)

Furthermore, in claiming that pairs of elements interact specifically/display super-additive activity, the authors rightfully state that, e.g., B+sp-kirrelLprm+endo16prm displays activity while B+endo16prm does not drive any reporter expression. However, given that sp-kirrelLprm+endo16prm itself drives expression, the activity of B+sp-kirrelLprm+endo16prm must be greater than the sum of the activities of B+endo16prm and sp-kirrelLprm+endo16prm in order to state that B and sp-kirrelLprm interact super-additively. In other words, the activity of kirrelLprm+endo16prm must be taken into account.Considering this activity, it does appear that B, C, E, F, H and I do act super-additively with Sp-kirrelLprm, considering at least one of the metrics of % of embryos expressing and % with PMC-specific expression (e.g. C+endo16prm drives GFP expression in 8.6% of embryos, Sp-kirrelLprm+endo16prm drives expression in 19%, and C+Sp-kirrelLprm+endo16prm drives expression in 37.8%, which is greater than 19+8.6). A+Sp-kirrelLprm+endo16prm and D+Sp-kirrelLprm+endo16prm, on the other hand, drive GFP expression both with less frequency and less PMC specificity than Sp-kirrelLprm+endo16prm, so I do not think it can be said that these elements interact specifically with kirrelLprm, and it seems that they do not have enhancer activity and may not be functional elements at all. If this is why BAC deletion constructs were not made for A and D, this should be explicitly stated.

We agree with the reviewer’s assessment and so never claim in the manuscript that A and D are functional elements. We have added a statement to the BAC results stating that deletions of A and D were not tested as there was no indication from the plasmid reporter analysis that they were functional elements.

It is interesting that several X+Sp-kirrelLprm+endo16prm combinations are more PMC-specific than Sp-kirrelLprm+endo16prm alone. This suggests that some of these elements may not only be enhancers, but also have silencer activity necessary to drive kirrelL expression in the proper spatial pattern. It is an interesting finding, in line with Gisselbrecht et al. Mol. Cell. (2020)'s finding that many silencers are enhancers in alternate tissue contexts, that should be highlighted.

This is an interesting point, but we hesitate to push it as the tandem arrangement of promoters in the *kirrelLprm+*endo16prm construct is unusual.

Finally, the interpretation in lines 269-271 that "the presence of the native Sp-kirrelL promoter mitigated the need for the C.ChIP element within element C to be adjacent to the promoter for strong PMC-specific GFP expression" (and restated in 274-276) is not strongly supported. Placing a spacer sequence with no enhancer or silencer activity in between C.ChIP and the promoter, and showing that this decreases activity as compared to when C.ChIP is adjacent to the promoter, would support this. However, the current data leave open the possibility that the part of element C downstream of C.ChIP contains sites for repressive factors, especially given that there is such a large difference in the activity of C.DNase and C.ATAC even though the former is only ~150 bp longer.

We hadn’t considered this interpretation and thank both Reviewers 1 and 3 for raising the point. To test whether the effect of deleting the region between C.ChIP and the promoter was due to the removal of repressor sites or to a change in the spacing between C.ChIP and the promoter (our original interpretation), we generated and tested a new construct that contained the region in question but in which the sequence of that region was randomly scrambled. We found that insertion of this sequence decreased activity compared to when C.ChIP was directly adjacent to the promoter. This strongly supports the view that the principle effect of deleting this region was to decrease the spacing between C.Chip and the promoter rather than removing repressor sites. These new data are shown in new Figure 4—figure supplement 3D,E. (Line 250-258)

kirrelL gene/protein tree:Ettensohn and Dey Dev. Biol. (2017) show a protein tree in which Sp-kirrelL, Lv-kirrelL and Pm-kirrelL cluster together within a set of transmembrane Ig-domain proteins, but I have not found such an analysis for the sequences of kirrelL proteins in the other species whose kirrelL cis-regulatory regions are tested for activity in this manuscript. This is necessary to confirm the identity of these kirrelL genes. If such an analysis has been published, that paper should be cited.

We have generated a new protein tree containing the additional echinoderm KirrelL protein sequences, all of which cluster convincingly with the sea urchin KirrelL proteins, and have included it as Figure 6—figure supplement 1.

Description of cross-species experiments:The sequences or coordinates of the putative kirrelL CREs from multiple species tested in reporter assays should be provided.

We have added this information in new Figure 6 – source data 2.

When describing the cross-species CRE reporter experiments (lines 376-381), please explicitly state the species for which the expression pattern of kirrelL is unknown, as this is relevant to the interpretation of these experiments.

We have added the following sentence: “To date, the embryonic expression of *kirrelL* has been examined in two sea urchins (*S. purpuratus* and *L. variegatus*) and a brittle star (*A. filiformis*) (Ettensohn and Dey, 2017; Dylus et al., 2018); in all three species, embryonic expression is restricted to skeletogenic mesenchyme cells.” (Line 316-319)

Also, the choice to examine the expression of kirrelL in the adult rudiment of L. variegatus rather than that of S. purpuratus needs to be explained.

We chose to work with *L. variegatus* here because this species can be raised through feeding larval stages much more quickly and easily than *S. purpuratus*. *KirrelL* expression and function has been characterized just as thoroughly in *L. variegatus* as in *S. purpuratus* (Ettensohn and Dey, 2017).

Finally, when presenting the results of the promoter swap experiment, I would explicitly state that PmG1 and PmG2 each confer a roughly similar increase in expression frequency and specificity to C+endo16prm as Sp-kirrelLprm does to more clearly convey that PmG1 and G2 can substitute for Sp-kirrelLprm in a meaningful way.

This statement has been added. (Line 375-377)

Reviewer #2:[…] Overall, this is a beautifully conducted study. The results are presented very clearly in both text and figures. Some specific questions and concerns follow below (numbers refer to line numbers in the manuscript):178. What is the justification for using the core promoter of a different gene for these experiments? Given that some enhancers show selectivity for nearby core promoters (at least in other systems), this seems like an odd choice.

In our initial experiments, we chose to use the *EpGFPII* reporter because this plasmid, originally developed in Eric Davidson’s lab, has been very widely used for cis-regulatory studies in sea urchins. EpGFPII was designed with a basal sea urchin core promoter (from the endo16 gene) that by itself does not drive appreciable reporter expression but that can be activated by many sea urchin enhancers. This reporter plasmid has proven very useful for comparing the activity of CREs in the context of a consistent and “neutral” core promoter element. On the other hand, the reviewer is absolutely correct that some enhancers show selectivity for promoters; for example, by tethering to specific elements in the proximal promoter region. As described in the paper, during the course of our work we did indeed uncover evidence of specific interactions between the endogenous *kirrelL* promoter region and distal CREs.

Figure 2B. I couldn't find any information about the number of replicates for these experiments, or any of the subsequent reporter assays presented in subsequent figures. A detailed tabulation for every separate experiment is not necessary, but a general statement about the number of replicates in a typical experiment would be very reassuring.

There’s a wide range here over the different constructs tested so it’s difficult to generalize. For constructs that were expressed very robustly and for which a single replicate yielded very large numbers of GFP-expressing embryos, we sometimes performed only a single trial. For constructs that yielded lower numbers of GFP-expressing embryos, however, multiple trials were needed to confirm a lack of expression or to produce enough GFP-expressing embryos to score a spatial expression pattern. Lastly, some constructs were used repeatedly as internal controls for the analysis of other constructs and so were injected 5-10 times. This is why in Figure 2 – source data 1 the number of injected embryos varies considerably from construct to construct. For what it’s worth, Figure 2 – source data 1 also shows that >20,000 injected embryos were scored during the course of this study.

196-198. The explicit criteria for defining "strong" and "weak" expression are helpful. That said, it wasn't immediately obvious from Figure 2C how these differ when looking at the images for ABC and C (weak) versus the other constructs (strong). The weak constructs look a bit out of focus but that could simply be the weaker signal. How consistent are these differences among embryos and from replicate to replicate?

The “weak PMC” and “strong PMC” designations are not based on our qualitative assessment of the intensity of the fluorescence. First, we scored as “expressing” every embryo that had any cells with detectable GFP (or mCherry) fluorescence. Unavoidably, this was limited by the sensitivity of our particular imaging system, but we used the same imaging system and parameters throughout our analysis (i.e., for all replicates). In addition, the scoring was all done by the same individual (J. Khor) who scored living embryos and was therefore able to focus through each specimen. It is important to note that the images in the paper show only single focal planes, and some cells are therefore out of focus. Next, all cells that exhibited detectable levels of fluorescence were scored for expression territory (PMC only, ectopic only, or PMC + ectopic), which provided a reliable estimate of the PMC-specificity of reporter expression. The determination of “weak” vs. “strong” PMC expression was based on the fraction of injected embryos that showed detectable levels of expression that was completely restricted to PMCs (no ectopic expression). If this value was <15% of injected embryos the construct was classified as “weak,” and if the value was >15%, the construct was classified as “strong”. Thus, our scoring did not require that we qualitatively guess at whether the intensity of the fluorescence was “weak” or “strong” in any individual embryo. We’re confident that variation in the intensity of the fluorescence was captured by our scoring system, as we found that constructs that produced qualitatively very faint fluorescence always fell into the bin of “weak PMC” expression, presumably because many embryos had low levels of expression that were below the detection limit of our imaging system.

212-214. Could this result also be explained by the presence of binding sites for repressors in ABC that BC.ATAC lacks?

We hadn’t considered this interpretation and thank both Reviewers 1 and 3 for raising the point. To test whether the effect of deleting the region between C.ChIP and the promoter was due to the removal of repressor sites or to a change in the spacing between C.ChIP and the promoter (our original interpretation), we generated and tested a new construct that contained the region in question but in which the sequence of that region was randomly scrambled. We found that insertion of this sequence decreased activity compared to when C.ChIP was directly adjacent to the promoter. This strongly supports the view that the principle effect of deleting this region was to decrease the spacing between C.Chip and the promoter rather than removing repressor sites. These new data are shown in new Figure 4—figure supplement 3. (Line 250-258)

249-250. What was the basis for considering this region to be the core promoter?

In the literature, core promoters are usually described as flanking the transcriptional start site by +/- 50 bp. We seem to have been a bit too generous here at ~300 bp and so have altered to text to read: “We found that a 301 bp region surrounding the transcriptional start site, a region we considered to include the *Sp-kirrelL* core promoter…”. (Line 219-221)

251-252. Are these features (strong, ubiquitous) true of the core promoter that was used for the experiments? If not, the earlier concern about choice of core promoter for the experiments is even more acute.

Please see our response to Point #2 of the Essential Revisions required by the Editor.

265-268. This is an interesting result. What does it imply mechanistically?

We discuss the evolutionary conservation of echinoderm *kirrelL* cis-regulatory CREs in the last section of the Discussion.

342. Why so much redundancy? This is touched on only briefly in the Discussion but seems like an important result.

We discuss this redundancy at length in the last paragraph of the section in the Discussion titled “The cis-regulatory apparatus of Sp-kirrelL.” We have also added a sentence citing a recent review on enhancer redundancy. (Line 457-459)

352-354. Is there any indication here (or in the previous results) of ectopic expression?More generally: The experiments do not seem geared to detect possible repressive functions for any of the regulatory elements. Are there reasons for thinking that repressive functions would not be needed? What keeps kirrelL from being transcribed outside of the single cell type where it is expressed?

Most of our analysis involved light microscopic observations of live transgenic embryos, which does allow us to detect ectopic expression and therefore to identify to CREs that might act to repress *kirrelL* expression in non-PMC territories. In general, we found no evidence of such repressive elements. We believe the cell type-specific expression of *kirrelL* is explained by the fact that two essential, positive regulators, Alx1 and Ets1, are co-expressed only in the PMC lineage and not in any other cells of the embryo. Alx1 is activated early in cleavage and is entirely restricted to the large micromere-PMC lineage throughout development. Ets1 is initially expressed specifically in the PMC lineage, although it is later expressed by a subset of non-skeletogenic mesoderm cells. Our mutational analysis of element C indicates that direct inputs from both Ets1 and Alx1 are required for activity (i.e., the two TFs act combinatorially, not redundantly), because mutation of either Alx1 half-site 2 or Ets1 site 1 abolishes the activity of the C element.

Figure 5: modifying the key in the lower left as follows might help make it easier to interpret panel C: GFP (deletion) mCherry (intact).

This change has been made and clarifies the figure.

376: Mention the extent of region tested here so that readers don't need to consult other parts of the paper to understand the basic experimental design. If not exactly the same region in all of the species, mention that, too.

We’ve added a statement here mentioning that we tested regions ~1-2 kb in size directly upstream of the translational start codon and have added a new table (Figure 6 – source data 2) that shows the precise genomic coordinates of the regions that were used. (Line 320-321)

383-384. The results support function being highly conserved. What about sequence? It looks like there is some patchy short sequence conservation between two sea urchins (Figure 1C) and two sea stars (Figure 7A). What about between these groups?

We thank the reviewer for recommending additional analysis here. The results have been added to the revised manuscript, as described in our response to the Editor’s letter (above).

In the literature "conserved regulatory element" more commonly applies to sequence than function. To avoid confusion, qualify "conserved" throughout the manuscript to clarify whether sequence or function is being discussed.

We thank the reviewer for catching this and have clarified our use of “conserved” throughout the manuscript.

447-448. Is there evidence for sequence conservation (and see above)?

Please see above.

622-623. Is this sentence intended to mean that specific binding sites are conserved? Or that binding sites for the same transcription factors are present? This is an important distinction. There is evidence from other systems that individual binding sites can turn over while conserving regulatory element function. That seems much more plausible in this case than super-strong conservation of tiny patches of sequence.

This sentence is intended to suggest that the same transcription factors likely mediate tethering between the C element and the promoter in both urchins and sea stars and that binding sites for these same proteins exist in both taxa. Of course, the order and spacing are probably very different and, as the reviewer points out, it is certainly possible that the specific sequences might have diverged to some extent, although they would still need to bind the same proteins and so would have to be at least partially conserved.

There are examples from other groups (especially vertebrates) where a regulatory element shows conserved function among highly divergent species. It would be helpful to mention this in the Discussion to provide some context. Is this the first reported case of conserved regulatory element function among deeply divergent echinoderms? If so, this is worth mentioning explicitly, again for context.

We mention the conservation of regulatory elements in other organisms in Paragraph 2 of the section on “Evolutionary conservation of echinoderm kirrelL CREs,” and cite a useful review by Rebeiz et al. Hinman et al. (2007) carried out cross-specific experiments and argued that the function of a module of Otx (OtxG) module is conserved in sea stars and sea urchins. Although we think our data are more thorough and compelling, we’ve added a sentence to the Discussion citing the Hinman paper and referring to our study as the second reported case.

Reviewer #3:[…] The paper is really well written and logically presented. If it were simply a cis regulatory story on a random gene I would have recommended a lower level journal. However, here the impressive conservation of the cis regulatory region elevates the paper quite significantly. Still, they don't show anything of the cis-regulatory region sequences, and these would help improve the paper – is the conservation actually small islands perhaps the Alx1 and Ets binding regions? Is it the basal promoter primarily? Or is the entire region fairly well conserved? The answer to those questions will be helped with the sequences. I point out that illustrating the protein sequences across the phylum is OK but is not really the focus of this paper.

We thank the reviewer for recommending additional analysis here. Additional information regarding CRE sequence conservation has been added to the revised manuscript, as described in our response to the Editor’s letter (above).

The other point that I think needs better quantification is the call that anything above 15% expression is considered "strong". I realize that these kind of reporter assays sometimes have a low percentage read-out, but the question is whether some elements are stronger than others. I ask simply the question about the strongest of the so-called strong vs the weakest of the so-called strong.

We agree with the reviewer that the binning we applied to the reporter data is an abstraction and almost certainly obscures some differences between constructs, but this kind of semi-quantitative scoring system was the only feasible option we could come up with given the large number of constructs we tested. We were concerned about making the assessment too granular and perhaps less reliable, so we chose to separate the constructs that showed PMC-specific expression into only two bins (“strong” and “weak”). The 15% cutoff was chosen arbitrarily, but whatever the threshold there will always be constructs that fall just above and below the boundary.

There are several components of the paper that would benefit from some revision to help me understand some of the interpretations.1. In Figure 1 showing ATAC-seq, DNAse hypersensitivity, ChIP seq data how did you call the peaks? Some, especially G, the promoter, are obvious but others are less so, and still others look similar to peaks you called as enhancers.

The ATAC-seq, DNase-seq, ChIP-seq,and eRNA peaks shown in Figure 1 were identified in previous published studies by Shashikant et al. (2018), Khor et al. (2019), and Khor et al., (2021), which are cited in the Figure 1 legend. In each case, the specific parameters used for peak-calling can be found in those publications.

2. You indicate that in scoring expression that greater than 15% of the embryos expressing a construct is strong expression. I understand some of the many reasons why 85% might not express. But, I would also like to know if some are stronger than others. For example is ABCDEFG stronger than DEFG and is that stronger than G? Also in Figure 2 I notice that C is indicated as weaker. The embryo illustrated has fewer fluorescent PMCs than other embryos in this panel with stronger promoters. Is it possible that element C somehow operates in a restricted number of cell bodies rather than all of them? After all, you show in other papers that there are mRNAs that are expressed in a restricted subset of PMCs even within the syncytium. Along the same line, although you indicate that Ets1 sites 2 and 3, when mutated, still allow for expression. Your images of those show stronger expression of GFP in the ventrolateral clusters than the spread of expression when all three Ets sites are control. Is that, or could that be meaningful?

With respect to the second part of the Reviewer’s comment, our microscopic analysis of transgenic embryos was carried out at 48 hpf, which is soon after PMC fusion is complete. Because GFP protein diffuses rapidly throughout the PMC syncytium, the entire PMC network is labeled in the embryos shown in Figures 1, 2, 6, S1-S5, and S7, despite the mosaic incorporation of transgenes in sea urchin embryos. This diffusion of GFP also means that possible regional variations in *kirrelL* expression during late development cannot be reliably detected using this reporter protein. The possibility that there is region-specific transcriptional regulation of *kirrelL* and other PMC effector genes at late embryonic stages is indeed very interesting to us and we are working on this problem separately, but this analysis will require the use of different (non-diffusible) reporter proteins.

Also, please note that our scoring did not require that we qualitatively estimate whether the intensity of the fluorescence in any individual embryo was “weak” or “strong”. We scored as positive every embryo that had any cells with detectable GFP (or mCherry) fluorescence. This was limited by the sensitivity of our particular imaging system, but we used the same imaging system and imaging parameters throughout our analysis (i.e., for all replicates). In addition, the scoring was all done by the same individual (J. Khor) who scored living embryos and was therefore able to focus through each specimen (note that the images in the paper show only single focal planes, and some cells are therefore out of focus). The determination of “weak” vs. “strong” PMC expression was based on the fraction of injected embryos that showed detectable levels of fluorescence (if this value was <15% of injected embryos the construct was classified as “weak,” and if the value was >15%, the construct was classified as “strong”). We’re confident that intensity of the fluorescence was captured by our scoring system, as we found that constructs that produced qualitatively faint fluorescence consistently fell into the bin of “weak” expression, presumably because many embryos had low levels of reporter protein that were below the detection limit of the imaging system. This qualitative scoring system definitely isn’t perfect, but the very large number of constructs we tested precluded us from more rigorously measuring the level of expression of each construct and we did use a quantitative approach (Nanostring) for the key BAC constructs.

3. The data in Figure 5 I find to be most valuable to the paper. That includes the nanostring data to indicate contributions relative to an mCherry control construct which is most informative.

No changes required.

4. The data on other species is most interesting. I see you have an alignment of the KirrelL proteins of the several species but that isn't really the story. You don't have anything to indicate the relative alignment of the same cis regulatory regions. The real question is how are those related? Are their islands of high conservation as seen earlier by Cameron and Davidson? Are there indels outside the areas that must be conserved enough to drive expression in S. purpuratus? It seems to me that these are the sequences of interest. Yes, there is demonstrated similarity in the KirrelL protein sequence but this paper is all about the cis regulation.

We thank the reviewer for recommending additional analysis here. Additional information regarding CRE sequence conservation has been added to the revised manuscript, as described in our response to the Editor’s letter.